# MACHINE UNLEARNING MEETS CONTINUOUS LEARNING: A THEORETICAL FOUNDATION

## ABSTRACT

Machine unlearning is designed to remove specific data from a trained model to protect privacy. However, a significant challenge arises in the field of continual learning, where models evolve without full access to past data due to ever-increasing storage burdens and environmental constraints. This is because current certified unlearning algorithms do not accommodate the complex model evolution in continual learning. In this work, we establish the first theoretical foundation connecting continual learning and machine unlearning, where the former aims to preserve knowledge across all previously trained tasks, while the latter requires efficient and immediate forgetting upon receiving unlearning requests. We successfully adapt two popular certified unlearning approaches, one leveraging gradients and the other Hessians, to function within a continual learning framework. We provide theoretical performance guarantees by analyzing two key metrics: the excess risk under continual learning and the unlearning loss. The combination of these two metrics jointly determines the final post-unlearning excess risk. Our analysis shows that our Hessian-based adaption algorithm largely outperforms the gradient-based algorithm, while the latter offers an advantage by reducing the storage cost to zero. We validate these theoretical findings with experiments on the MNIST dataset, which also demonstrate the effect of the sequence of unlearning requests.

## 1 INTRODUCTION

As machine learning applications ranging from large language models to healthcare systems increasingly rely on user data to deliver personalized services, concerns over privacy have grown substantially. To uphold users' right to data deletion and protect users' privacy, the field of machine unlearning has advanced rapidly recently, developing methods to remove data influence from the trained model without the costly process of retraining models from scratch (Gupta et al. (2021); Warnecke et al. (2021); Sekhari et al. (2021); Suriyakumar & Wilson (2022); Chien et al. (2022; 2024a); Qiao et al. (2024)). Within this landscape, $(\varepsilon, \delta)$-certified unlearning (Guo et al. (2019)), has emerged as a promising direction, providing rigorous indistinguishability theoretical guarantees relative to perfect retraining. The goal of an $(\varepsilon, \delta)$-certified unlearning algorithm is to approximate perfect retraining on the remaining data and ensure the $(\varepsilon, \delta)$-guarantee via a noise mechanism.

The current certified unlearning approaches can be broadly categorized into two directions. Hessian-based methods extract Hessian information from the dataset and apply a second-order approximation to estimate the retrained model (Sekhari et al. (2021); Suriyakumar & Wilson (2022); Liu et al. (2023); Qiao et al. (2024); Basaran et al. (2025)). Gradient-based methods (Neel et al. (2021); Chien et al. (2024a;b); Koloskova et al. (2025)) instead continue updating on the remaining data (excluding the data points to be unlearned), using carefully designed update rules and noise mechanisms to ensure forgetting (Kirkpatrick et al. (2017)). However, most existing certified unlearning algorithms are developed under the assumption that the system always has access to the full dataset from which unlearning requests originate (Neel et al. (2021); Qiao et al. (2024)), or are designed for one-time unlearning in simple learning frameworks (Sekhari et al. (2021); Basaran et al. (2025)).

Modern machine learning frameworks, such as ChatGPT, are increasingly adopting a continual learning approach, where models are trained on tasks as they arrive sequentially. Because of this shift, it is now crucial to apply machine unlearning in this continual learning environment. Since storing past datasets for retraining is often infeasible in continual learning due to storage or en-

Figure 1: Two-stage continual learning and unlearning at time $t$: starting from the last model $w_{t-1}^{-\mathbf{S}_{1:t-1}}$ at time $t-1$, we first train on task $t$ with dataset $D_t$ to obtain $w_t^{-\mathbf{S}_{1:t-1}}$ in Stage I. Upon receiving a possible deletion request $S_t$, in Stage II, the unlearning scheme $\mathcal{R}_{\mathcal{A}}(\cdot, D_{1:t}, \mathbf{S}_{1:t})$ in (2) updates the internal model $w_t^{-\mathbf{S}_{1:t}}$, and publishes the final unlearned model $\tilde{w}_t^{-\mathbf{S}_{1:t}}$, by noise adding mapping $f$ in (3) to achieve $(\varepsilon, \delta)$-certified continual unlearning in Definition 2.1. Then, it feeds $w_t^{-\mathbf{S}_{1:t}}$ to train on next task $t+1$.

vironmental constraints, the problem in continual learning is that models frequently suffer from catastrophic forgetting of previously acquired knowledge, which severely degrades generalization performance (Swartworth et al. (2023); Evron et al. (2022); Deng et al. (2025)).

When machine unlearning meets continual learning, two new challenges emerge. First, unlearning must be performed on data from tasks trained several rounds ago. This presents a challenge because, in continual learning, the model evolves in a more complex and dynamic manner with non-i.i.d. tasks, while the original datasets are no longer accessible. As a result, current certified unlearning algorithms cannot function (Qiao et al. (2024); Neel et al. (2021); Basaran et al. (2025)). Second, we need to analyze and balance **unlearning loss** against the **excess risk of continual learning** to jointly minimize the final post-unlearning excess risk. This analysis must also account for additional storage costs and requires a thorough understanding of how the unlearning request pattern affects the continual learning process. Note that only a few work have begun investigating continual learning and unlearning, yet most are system works with heuristics-based designs for ad hoc cases, without providing any theoretical performance guarantee (Liu et al. (2022); Chatterjee et al. (2024); Cha et al. (2024); Huang et al. (2025)). More discussion of related work is introduced in Appendix A.

Our main contributions are summarized as follows.

- We present the first theoretical investigation into the problem of unlearning within a continual learning framework, which we term continual learning-unlearning. Our research establishes a novel analytical connection, bridging the excess risk introduced by continual learning and the unlearning loss resulting from unlearning. To begin, we prove an excess-risk bound for the $\ell_2$-regularized continual learning algorithm, extending prior results from linear to nonlinear convex models.

- We adapt the existing gradient-based certified unlearning that leverages the forgetting effect of continual learning, directly adding noise to achieve certified unlearning. We provide performance guarantees on its unlearning loss, which, together with the excess risk from continual learning, jointly determine the final post-unlearning excess risk.

- We adapt Hessian-based unlearning approaches to the continual learning framework, which achieve lower unlearning loss than gradient-based methods but incur additional storage costs. We further show that their unlearning loss depends critically on the order of requests, offering insights into regulating request arrivals. Moreover, we incorporate the natural forgetting effect to improve the Hessian-based unlearning to reduce storage costs. These theoretical findings are validated by experiments on MNIST, which also highlight the impact of the unlearning sequence.

## 2 PROBLEM FORMULATION FOR CONTINUAL LEARNING-UNLEARNING

In this section, we mathematically model the machine unlearning in the continual learning process over two stages at any time $t$ in Fig. 1. We then formulate $(\varepsilon, \delta)$-certified continual unlearning problem, the unlearning loss objective in the unlearning process and excess risk objective in continual learning.

### 2.1 CONTINUAL LEARNING MODEL IN STAGE I

In the continual learning, tasks arrive sequentially in the discrete-time horizon of $t = 1, \ldots, T$. Each task $t$ provides a dataset $D_t = \{z_i\}_{i=1}^{n_t} \in \mathcal{Z}^{n_t}$, where $\mathcal{Z}^{n_t}$ is the data instance space of sample

size $n_t$. Denote $D_{1:t} = (D_1, \ldots, D_t)$ for past dataset history. Let $\ell(w, z)$ be the loss function of any model parameter $w$ in space $\mathcal{W} \subseteq \mathbb{R}^d$ of $d$ dimensions on instance $z$ at any time. For analysis tractability, we adopt the following standard assumption on this loss function in the unlearning literature (e.g., Sekhari et al. (2021); Suriyakumar & Wilson (2022); Qiao et al. (2024)).

**Assumption 2.1.** *The loss function $\ell$ is $L$-Lipschitz continuous, $\mu$-strongly convex, and $M$-smooth.*

We study the following widely used $\ell_2$-regularized continual learning algorithm below.

- $\ell_2$-CL$(w_{t-1}, D_t)$ with inputs of a model $w_{t-1}$ and task dataset $D_t$:

$$w_t = \arg\min_w \sum_{z_i \in D_t} \ell(w, z_i) + \lambda/2 \|w - w_{t-1}\|_2^2. \tag{1}$$

   Return $w_t$

(1) updates the model $w_t$ through task $t$ by minimizing the loss on dataset $D_t$ together with a regularization term that penalizes the distance from the previous model $w_{t-1}$. This method mitigates forgetting and has shown practical effectiveness (Kirkpatrick et al. (2017)). Its tractable structure has also made it central to recent theoretical studies of continual learning (e.g., Zhao et al. (2024)). We therefore initiate the study of certified unlearning in continual learning under this method.

We formalize the continual learning algorithm as the following mapping from any sequence of past datasets $D_{1:t}$ to the model $w_t$: $\mathcal{A} : \mathcal{Z}^{n_1} \times \cdots \times \mathcal{Z}^{n_t} \to \mathcal{W}$. The weight $\lambda$ in (1) controls the degree of forgetting of prior knowledge. Later, we will show that the choice of hyperparameter $\lambda$ plays a crucial role in both continual learning and unlearning.

## 2.2 Certified unlearning model in continual learning process in Stage II

We study a continuous task-level unlearning [1], which may arise at any time during the continual learning process. As illustrated in Fig. 1, after finishing training on task $t$ with data $D_t$ by (1) in Stage I, the system may receive a deletion request $S_t$ to delete all the data samples in $D_i$ of any prior task $i \in S_t$. $S_t$ is a subset of task indices that have been trained but have not yet been deleted, i.e., $S_t \subseteq \{1, \ldots, t\} \setminus S_{\leq t}$, with $S_{\leq t} = \cup_{i=1}^{t-1} S_i$, $S_0 = \emptyset$, where $S_{\leq t}$ is the accumulated set of tasks to delete up to time $t$. Let $\mathbf{S}_{1:t} = (S_1, \ldots, S_t)$ denote the sequence of deletion requests issued by time $t$.

We use the model $w_t^{-\mathbf{S}_{1:t}}$ to denote the instance trained on tasks $D_{1:t}$ and unlearned on requests $\mathbf{S}_{1:t}$. Starting from last model $w_{t-1}^{-\mathbf{S}_{1:t-1}}$ on the left side of Fig. 1, we first train on the new task $D_t$ to obtain $w_t^{-\mathbf{S}_{1:t-1}}$ (subscript incremented), then apply the unlearning algorithm $\mathcal{R}_\mathcal{A}(w_t^{-\mathbf{S}_{1:t-1}}; D_{1:t}, \mathbf{S}_{1:t})$ to unlearn tasks in $S_t$, yielding $w_t^{-\mathbf{S}_{1:t}}$ (superscript incremented), which serves as the internal state for subsequent differential-privacy enforcement.

Formally, we define the unlearning algorithm as a mapping to the model parameter space $\mathcal{W}$:

$$\mathcal{R}_\mathcal{A} : \mathcal{W} \times \left(\mathcal{Z}^{n_1} \times \cdots \times \mathcal{Z}^{n_t}\right) \times \mathcal{S}_t \longrightarrow \mathcal{W}, \tag{2}$$

where $\mathcal{S}_t := \left\{ (S_1, \ldots, S_t) : S_k \subseteq \{1, \ldots, k\} \setminus \bigcup_{i<k} S_i, \ k = 1, \ldots, t \right\}$ is the set of all possible deletion-request histories. We then apply a public noise-adding mapping to the result $w_t^{-\mathbf{S}_{1:t}}$ of $\mathcal{R}_\mathcal{A}$:

$$f : \mathcal{W} \to \mathcal{W} \tag{3}$$

to output $\tilde{w}_t^{-\mathbf{S}_{1:t}} = f\left(w_t^{-\mathbf{S}_{1:t}}\right)$ and release it to the public (e.g., AWS Marketplace). The model $w_t^{-\mathbf{S}_{1:t}}$ is then carried over to time $t+1$ and trained on task $t+1$, repeating two learning and unlearning stages above.

## 2.3 Problem formulation with continual learning and unlearning objectives

In machine unlearning, given the sequence of unlearning requests $\mathbf{S}_{1:t}$, the goal of unlearning output $w_t^{-\mathbf{S}_{1:t}} = \mathcal{R}_\mathcal{A}(w_t^{-\mathbf{S}_{1:t-1}}, \mathbf{S}_{1:t}, D_{1:t})$ is to approximate the perfect retraining model $w_t^{-S_{\leq t}} = \mathcal{A}((D_i)_{i \in [t] \setminus S_{\leq t}})$ obtained by continual learning mapping $\mathcal{A}$ on the remaining tasks sequence $\{1, \ldots, t\} \setminus S_{\leq t}$. While the retraining model depends only on the remaining datasets, the unlearning mapping $\mathcal{R}_\mathcal{A}$ depends on the entire history of deletion requests. We then require the

---

[1] Although this paper focuses on unlearning of a whole task's dataset, we can easily extend our model and method to delete individual data samples within any task.

public model $\tilde{w}_t^{-\mathbf{S}_{1:t}} = f(w_t^{-\mathbf{S}_{1:t}})$ to be indistinguishable from the perfect retraining model $w_t^{-S_{\leq t}}$ in terms of their output probabilities. We develop an $(\varepsilon, \delta)$-certified continual unlearning requirement by extending the standard $(\varepsilon, \delta)$-certified unlearning definition from non-continual learning framework (Neel et al. (2021)).

**Definition 2.1.** $(\varepsilon, \delta)$-*certified continual unlearning. Let* $0 < \varepsilon \leq 1$ *and* $\delta > 0$. *For every measurable parameter space* $\mathcal{W} \subseteq \mathbb{R}^d$, *an unlearning algorithm* $\mathcal{R}_{\mathcal{A}}$ *is said to satisfy* $(\varepsilon, \delta)$-*certified continual unlearning if, for every time* $t$,

$$\Pr\big(\tilde{w}_t^{-\mathbf{S}_{1:t}} \in \mathcal{W}\big) \leq e^\varepsilon \Pr\big(w_t^{-S_{\leq t}} \in \mathcal{W}\big) + \delta, \ \Pr\big(w_t^{-S_{\leq t}} \in \mathcal{W}\big) \leq e^\varepsilon \Pr\big(\tilde{w}_t^{-\mathbf{S}_{1:t}} \in \mathcal{W}\big) + \delta. \ (4)$$

Under the $(\varepsilon, \delta)$-certified unlearning constraint, the objective of continual learning–unlearning is to minimize the excess risk on all the tasks, as defined below.

**Definition 2.2.** *Post-unlearning excess risk. Let* $\mathcal{D}_t$ *denote the data-generating distribution for task* $t$, *from which the dataset* $D_t$ *is drawn. Define the task-wise population loss as* $F_\tau(w) = \mathbb{E}_{z \sim \mathcal{D}_t}[\ell(w, z)]$ *for task* $\tau$. *Then, the excess risk that quantifies the performance of any continual learning unlearning algorithm at time* $t$ *is defined below:*

$$\mathbb{E}\big[\tfrac{1}{t-|S_{\leq t}|} \sum_{\tau=1, \tau \notin S_{\leq t}}^t F_\tau(\tilde{w}_t^{-\mathbf{S}_{1:t}})\big] - \min_w \tfrac{1}{t-|S_{\leq t}|} \sum_{\tau=1, \tau \notin S_{\leq t}}^t F_\tau(w), \quad (5)$$

*where the randomness is over the per-task datasets* $D_t$ *used to train* $w_t^{-\mathbf{S}_{1:t}}$.

The post-unlearning excess risk in equation 5 can be rewritten as the sum of the following two parts:

$$\underbrace{\mathbb{E}\left[\frac{1}{t-|S_{\leq t}|} \sum_{\tau=1, \tau \notin S_{\leq t}}^t F_\tau(\tilde{w}_t^{-\mathbf{S}_{1:t}}) - F_\tau(w_t^{-S_{\leq t}})\right]}_{\text{unlearning loss in machine unlearning}} \quad (6)$$

$$+ \underbrace{\mathbb{E}\left[\frac{1}{t-|S_{\leq t}|} \sum_{\tau=1, \tau \notin S_{\leq t}}^t F_\tau(w_t^{-S_{\leq t}})\right] - \frac{1}{t-|S_{\leq t}|} \min_w \sum_{\tau=1, \tau \notin S_{\leq t}}^t F_\tau(w),}_{\text{excess risk in continual learning}} \quad (7)$$

where (6) is the unlearning loss induced by the unlearning algorithm, and (7) is the excess risk arising from the continual learning algorithm. Since the unlearning loss depends on $\|w_t^{-\mathbf{S}_{1:t}} - w_t^{-S_{\leq t}}\|$, we also define this quantity as the approximation error, which will be discussed later.

Importantly, selecting a continual learning algorithm $\mathcal{A}$ that prevents forgetting to minimize the excess risk in (7) may inversely increase unlearning loss (6), in sharp contrast to prior certified unlearning work (Sekhari et al. (2021); Suriyakumar & Wilson (2022); Liu et al. (2023)). Thus, our continual learning-unlearning introduces a new challenge: balancing between minimzing the excess risk during continual learning and the unlearning loss under machine unlearning.

## 3 OUR PRELIMINARY RESULTS ON EXCESS LOSS IN CONTINUAL LEARNING

In this section, before studying unlearning, we present new theoretical results establishing an upper bound on the excess risk in continual learning for (7) under algorithm (1), which serves as building blocks for the following sections' analysis on post-unlearning excess risk.

**Theorem 3.1.** *Denote* $w_t^* = \arg\min_w F_t(w)$ *as the optimal model that minimizes the population loss* $F_t(w) = \mathbb{E}_{z \sim D_t}[\ell(w, z)]$ *of task* $t$. *For the retraining sequence* $\{1, \ldots, t\} \setminus S_{\leq t}$, *let* $\{\tau_1, \ldots, \tau_k\} = \{1, \ldots, t\} \setminus S_{\leq t}$, *where* $k = t - |S_{\leq t}|$. *Then, for model* $w_t^{-S_{\leq t}} = w_{\tau_k}^{-S_{\leq \tau_k}}$ *obtained by training with (1) under the sequence* $\{\tau_1, \ldots, \tau_k\}$, *the excess risk in (7) is upper bounded by* $\mathcal{E}^{-S_{\leq t}}(\lambda)$:

$$\mathcal{E}^{-S_{\leq t}}(\lambda) := \frac{LM}{M+\lambda}\left(\sum_{i=1}^k \sum_{j=2, j\neq i}^k \rho^{\tau_k - \tau_j} \|w_{\tau_j}^* - w_{\tau_i}^*\| + \rho^{\tau_k - 1} \sum_{i=1}^k \|w_{\tau_i}^*\|\right) + L\rho^{\tau_k} \sum_{i=2}^k \|w_{\tau_1}^* - w_{\tau_i}^*\|$$

$$+ \frac{2\sqrt{2}LM}{\mu(M+\lambda)}\left(\frac{1-\rho^{\tau_k}}{|D_{\tau_1}|(1-\rho)} + \sum_{i=2}^k \frac{1}{|D_{\tau_i}|}\left((\tau_k - 2)\rho^{\tau_k - \tau_i} + \rho^{\tau_k - 1} + \frac{1-\rho^{\tau_k}}{1-\rho}\right)\right)$$

$$+ \frac{2\sqrt{2}L^2\rho^{\tau_k}}{\mu}\left(\frac{(\tau_k - 1)}{\sqrt{|D_{\tau_1}|}} + \sum_{i=2}^k \frac{1}{\sqrt{|D_{\tau_i}|}}\right) + \sum_{i=1}^k \frac{4L^2}{\mu|D_{\tau_i}|}, \quad (8)$$

*where $\rho := \frac{\lambda}{\mu+\lambda}$.*

The proof of Theorem 3.1 is given in Appendix B.1. This result greatly extends the generalization loss analysis from the existing linear model (e.g., Lin et al. (2023)). The choice of weight $\lambda$ that minimizes the upper bound $\mathcal{E}^{-S_{\leq t}}(\lambda)$ depends on the difference between tasks $i$ and $j$ (i.e., $\|w_i^* - w_j^*\|$) and the norm of each optimal model $\|w_i^*\|$. This upper bound does not vanish even with sufficiently large sample sizes $|D_i|$ for each task $i = 1, \ldots, t$, due to model heterogeneity and the non-i.i.d. nature of the dataset. In most cases a positive $\lambda$ is needed to achieve a smaller bound, where we discuss in Appendix B.2. Though continual learning generally benefits from a nonzero $\lambda$ to control forgetting, we will later see that this advantage does not extend to unlearning.

## 4 NATURAL FORGETTING CONTINUAL UNLEARNING ALGORITHM

In this section, we analyze and leverage the natural forgetting effect in continual learning to adapt gradient-based unlearning methods (Neel et al. (2021)) to the continual learning setting, achieving an $(\varepsilon, \delta)$-certified continual unlearning guarantee.

Note that models sequentially updated with new tasks by algorithm (1) and forget knowledge of past tasks naturally resemble the property of gradient-based unlearning methods (Neel et al. (2021); Chien et al. (2024a;b)). However, this phenomenon cannot be overly exploited, as excessive updates and noise added on new tasks cause the model to forget all previously acquired knowledge, not just the tasks requested for unlearning, which is precisely what standard continual learning seeks to prevent. Therefore, we only leverage the natural forgetting of model $w_t$ to design noise that masks the difference between $w_t$ and $w_t^{-S_{\leq t}}$, without further adapting gradient-based methods, which inherently forget all past tasks together.

---

**Algorithm 1:** Natural forgetting continual unlearning algorithm

1 Initialize $S_{\leq 0} = \emptyset$, $S_{1:1} = \emptyset$;
2 **for** $t \leftarrow 1$ **to** $T$ **do**
3     **Stage I. continual learning on task** $t$:
4     Receive dataset $\{z_i\}_{i \in D_t}$, $w_t \leftarrow \ell_2\text{-CL}(w_{t-1}, \{z_i\}_{i \in D_t})$ in (1);
5     **Stage II. unlearning and publishing model**:
6     Receive deletion request $S_t \subseteq [t] \setminus S_{\leq t-1}$, update $S_{\leq t} = S_t \cup S_{\leq t-1}$, $\mathbf{S}_{1:t} = (\mathbf{S}_{1:t-1}, S_t)$;
7     **if** $S_{\leq t} = \emptyset$; /* Enters whenever we update unlearning requests history. */
8     **then**
9         $\tilde{w}_t^{-\mathbf{S}_{1:t}} \leftarrow w_t$;
10     **else**
11         Draw $\epsilon_t \sim \mathcal{N}(\mathbf{0}, \sigma\mathbb{I})$, $\sigma = \gamma_t(\mathbf{S}_{1:t})\sqrt{2\ln(1.25/\delta)}/\varepsilon$ where $\gamma_t(\mathbf{S}_{1:t})$ is given by (9);
12         **Output:** $\tilde{w}_t^{-\mathbf{S}_{1:t}} \leftarrow w_{t,0} + \epsilon_t$;

---

Therefore, we skip the $\mathcal{R}_\mathcal{A}(\cdot, D_{1:t}, \mathbf{S}_{1:t})$ step in Stage II of Fig. 1 and directly design the noise mapping $f$ for certified unlearning, which offers the simplest implementation and requires no additional storage. The proposed natural forgetting-continual learning algorithm is presented in Alg. 1. In its Stage I, we keep using (1) to train on each new task. In Stage II, if $S_{\leq t}$ contains any unlearning request up to time $t$, we inject noise in line 12 to mask the gap between $w_t$ and $w_t^{-S_{\leq t}}$.

A standard result following Dwork et al. (2014) and Qiao et al. (2024) is that adding noise with standard deviation $\|w_t^{-S_{\leq t}} - w_t\|\sqrt{2\ln(1.25/\delta)}/\varepsilon$ on $w_t$ to obtain $\tilde{w}_t^{-\mathbf{S}_{1:t}}$ can ensure $(\varepsilon, \delta)$-certified unlearning with $\tilde{w}_t^{-\mathbf{S}_{1:t}}$. We just need to establish the upper bound $\gamma_t(\mathbf{S}_{1:t})$ on $\|w_t^{-S_{\leq t}} - w_t\|$ for the noise design, leveraging the forgetting effect inherent in the $\ell_2$-regularized continual learning algorithm (1), together with the performance guarantees of Alg. 1 in Theorem 4.1.

**Theorem 4.1.** *For each $i$-th unlearning request $t_i \in U_t = \{t_1, \ldots, t_k\}$, let $n_{t_i, s+1}^i$ denote the number of tasks in time interval $[s+1, t_i]$ that have been deleted by time $t_i$. Alg. 1 using approximation error's upper bound $\gamma_t(\mathbf{S}_{1:t})$ below tells unlearning loss and guarantees $(\varepsilon, \delta)$-certified continual unlearning in Definition 2.1:*

$$\|w_t^{-S_{\leq t}} - w_t\| \leq \gamma_t(\mathbf{S}_{1:t}) = \frac{L}{\lambda} \sum_{i=1}^{k} \sum_{s \in S_{t_i}} \rho^{t-s-n_{t,s+1}^k}. \tag{9}$$

*Further, the output model $\tilde{w}_t^{-\mathbf{S}_{1:t}}$ of Alg. 1 achieves the following post-unlearning excess risk upper bound as defined in Definition 2.2:*

$$L\left(\frac{\sqrt{2d\ln(\frac{1.25}{\delta})}}{\varepsilon} + 1\right)\gamma_t(\mathbf{S}_{1:t}) + \mathcal{E}^{-S_{\leq t}}(\lambda), \tag{10}$$

*where $\mathcal{E}^{-S_{\leq t}}(\lambda)$ is given in (8), and $\gamma_t(\mathbf{S}_{1:t})$ is in (9).*

The proof of Theorem 4.1 is given in Appendix C.1. Theorem 4.1 shows the beneficial forgetting effect that each unlearned task $s$ contributes an error term proportional to $\rho^{t-s-n_{t,s+1}^k}\frac{L}{\lambda}$ to (9). Since $\rho < 1$, the larger the number of tasks remaining from task $s$ to the current time $t$, the smaller the unlearning loss attributable to task $s$. Moreover, the unlearning loss' upper bound $\gamma_t(\mathbf{S}_{1:t})$ approaches zero for $\lambda = 0$ and $\lambda \to \infty$.

Though the system's published model ensures the certified unlearning, Alg. 1 internally maintains the secret model $w_t$ for future continual learning on task $t+1$, which may still contain information from all deleted tasks. We extend Alg. 1 to ensure stronger certified unlearning in Appendix C.2

Alg. 1 yields a small unlearning loss (6) when forgetting early tasks with small $\rho$, and only requires zero storage overhead. However, it may not guarantee a uniformly small post-unlearning excess risk to unlearn recent tasks. Thus, in the next section, we adapt the Hessian-based unlearning algorithm to the continual learning setting at the cost of extra storage deployment, which more accurately approximates the retraining model.

## 5 HESSIAN-BASED CONTINUAL UNLEARNING ALGORITHM

In this section, we adapt the Hessian-based unlearning methods (e.g., Sekhari et al. (2021); Suriyakumar & Wilson (2022); Qiao et al. (2024)) to continual learning.

### 5.1 HESSIAN-BASED ALGORITHM DESIGN FOR CONTINUAL LEARNING-UNLEARNING

We first provide the adaptation idea behind our Hessian-based unlearning algorithm. Consider two unlearning requests arrive at time $t_k$ and $t_{k-1}$, with no requests in between, such that $w_{t_k}^{-\mathbf{S}_{1:t_k-1}} = w_{t_k}^{-\mathbf{S}_{1:t_{k-1}}}$ as defined in Section 2. Our goal is to approximate the new retraining model $w_{t_k}^{-S_{\leq t_k}} = w_{t_k}^{-S_{\leq t_{k-1}} \cup S_{t_k}}$ from $w_{t_k}^{-\mathbf{S}_{1:t_k-1}}$. Denote the empirical loss on each task $i$ as $\tilde{F}_i(w) = \frac{1}{|D_t|}\sum_{z_i \in D_t}\ell(w, z_i)$. Under the continual learning algorithm (1), we have the following first-order conditions:

$$\nabla\tilde{F}_{t_k}(w_{t_k}^{-S_{\leq t_k}}) + \lambda(w_{t_k}^{-S_{\leq t_k}} - w_{t_k-1}^{-S_{\leq t_k}}) = 0, \ \nabla\tilde{F}_{t_k}(w_{t_k}^{-\mathbf{S}_{1:t_k-1}}) + \lambda(w_{t_k}^{-\mathbf{S}_{1:t_k-1}} - w_{t_k-1}^{-\mathbf{S}_{1:t_k-1}}) = 0.$$

Having Taylor expansion of $\nabla F_{t_k}(w_{t_k}^{-S_{\leq t_k}})$ at the point of $w_{t_k}^{-\mathbf{S}_{1:t_k-1}}$, we have

$$w_{t_k}^{-S_{\leq t_k}} - w_{t_k}^{-\mathbf{S}_{1:t_k-1}} \approx \left(\nabla^2\tilde{F}_{t_k}(w_{t_k}^{-\mathbf{S}_{1:t_k-1}}) + \lambda I\right)^{-1}\lambda\left(w_{t_k-1}^{-S_{\leq t_k}} - w_{t_k-1}^{-\mathbf{S}_{1:t_k-1}}\right).$$

Suppose task $t_k - 1 \in S_t$ requests to unlearn at time $t_k$. Since task $t_k - 1$ is removed when retraining on all the remaining tasks, we have $w_{t-1}^{-S_{\leq t}} = w_{t-2}^{-S_{\leq t}\setminus\{t-1\}}$ and can rewrite the Taylor expansion above as:

$$w_{t_k}^{-S_{\leq t_k}} \approx \left(\nabla^2\tilde{F}_{t_k}(w_{t_k}^{-\mathbf{S}_{1:t_k-1}}) + \lambda I\right)^{-1}\lambda\left(w_{t_k-2}^{-S_{\leq t_k}\setminus t-1} - w_{t_k-2}^{-\mathbf{S}_{1:t_k-1}}\right) \tag{11}$$

$$+ \left(\nabla^2\tilde{F}_t(w_{t_k}^{-\mathbf{S}_{1:t_k-1}}) + \lambda I\right)^{-1}\lambda\Delta_{t_k-1} + w_{t_k}^{-\mathbf{S}_{1:t_k-1}}, \tag{12}$$

where $\Delta_{t_k-1} = w_{t_k-2}^{-\mathbf{S}_{1:t_k-1}} - w_{t_k-1}^{-\mathbf{S}_{1:t_k-1}}$ represents the model update upon training on task $t_k - 1$.

Forward iterating the first term of (11) is challenging due to the disruptive unlearning task pattern. The tasks in the set $S_{\leq t}\setminus\{\tau, \ldots, t\}$ differ from those in the sequence $\mathbf{S}_{1:\tau}$ under arbitrary unlearning sequence, making the forward iterations highly heterogeneous. In particular, if a request arriving at time $t_k$ is to unlearn task before $t_{k-1}$, it disrupts the previous unlearning sequence $\mathbf{S}_{1:t'}$ such that the model $w_{t_k}^{-\mathbf{S}_{1:t_k-1}}$ may embed the information of $s$ in a complex manner. To address this challenge, we propose the unlearning update in line 10 of Alg. 2, which robustly achieves an exact second-order approximation to the retrained model for any unlearning sequence.

In Stage II of unlearning in Alg. 2, when a request $S_t$ arrives, the model is updated from $w_t^{-\mathbf{S}_{1:t-1}}$ to $w_t^{-\mathbf{S}_{1:t}}$ using the correction $\bar{\Delta}_t$ in (13). The first term of $\bar{\Delta}_t$ (13) removes the tasks newly re-

---

**Algorithm 2:** Hessian-based continual unlearning algorithm

---

1   Initialize $U_t = \emptyset$, $S_{\leq 0} = \emptyset$, $S_{1:1} = \emptyset$;

2   **for** $t \leftarrow 1$ **to** $T$ **do**

3     **Stage I: learning and precomputation on task** $t$:

4     Receive dataset $\{z_i\}_{i \in D_t}$, $w_t^{-\mathbf{S}_{1:t-1}} \leftarrow \ell_2\text{-CL}\,(\,w_{t-1}^{-\mathbf{S}_{1:t-1}},\, D_t)$ in Alg .1;

5     $\Delta_t \leftarrow w_{t-1}^{-\mathbf{S}_{1:t-1}} - w_t^{-\mathbf{S}_{1:t-1}}$, $H_t \leftarrow \frac{1}{|D_t|} \sum_{z_i \in D_t} \nabla^2 \ell(w_t^{-\mathbf{S}_{1:t-1}}, z_i)$, store $H_t$ and $\Delta_t$;

6     **Stage II: system unlearning**:

7     Receive deletion request $S_t \subseteq [t] \setminus S_{\leq t-1}$;

8     **if** $S_t \neq \emptyset$ **then**

9       Update $S_{\leq t} = S_t \cup S_{\leq t-1}$, $\mathbf{S}_{1:t} = (\mathbf{S}_{1:t-1}, S_t)$, $U_t = \{t\} \cup U_{t-1}$, $w_t^{-\mathbf{S}_{1:t}} = w_t^{-\mathbf{S}_{1:t-1}} + \bar{\Delta}_t$,
       where

$$\bar{\Delta}_t = \sum_{s \in S_t} \big( \prod_{i=s+1, i \notin S_t}^{t} (H_i + \lambda I)^{-1} \lambda I \big) \Delta_s +$$

$$\sum_{s \in S_{\leq t-1}} \big( \prod_{i=s+1, i \notin S_{\leq t}}^{t} (H_i + \lambda I)^{-1} \lambda I \big) \Delta_s - \sum_{\tau \in U_{t-1}} \big( \prod_{i=\tau+1, i \notin S_{\leq t}}^{t} (H_i + \lambda I)^{-1} \lambda I \big) \bar{\Delta}_\tau \quad (13)$$

10     **else**

11       Update $S_{\leq t} = S_{\leq t-1}$, $\mathbf{S}_{1:t} = \mathbf{S}_{1:t-1}$, $U_t = U_{t-1}$;

12     **publishing model**:

13     **if** $S_{\leq t} \neq \emptyset$ **then**

14       Draw $\epsilon_t \sim \mathcal{N}\big(\mathbf{0}, \sigma\mathbb{I}\big)$, $\sigma = \gamma_t(\mathbf{S}_{1:t})\, \sqrt{2\ln(1.25/\delta)}/\varepsilon$ with $\gamma_t$ in (14);

15       **Output**: $\tilde{w}_t^{-\mathbf{S}_{1:t}} \leftarrow w_t^{-\mathbf{S}_{1:t}} + \epsilon_t$;

16     **else**

17       **Output**: $\tilde{w}_t^{-\mathbf{S}_{1:t}} \leftarrow w_t^{-\mathbf{S}_{1:t}}$;

---

quested at time $t$, while the second term in bracket adjusts for handling interference with previously unlearned tasks in $S_{\leq t-1}$ and earlier corrections $\bar{\Delta}_\tau$ with $\tau < t$.

## 5.2   PERFORMANCE ANALYSIS OF POST-UNLEARNING EXCESS RISK

We first analyze the approximation error $\|w_t^{-S_{\leq t}} - w_t^{-\mathbf{S}_{1:t}}\|$ incurred by the unlearning in line 10 of Alg. 2, which determines the noise required to achieve $(\varepsilon, \delta)$-certified unlearning in line 15 of Alg. 2, and the post-unlearning excess risk. Proposition 5.1 presents the first-order upper bound on the approximation error, and we further prove a second-order upper bound in Proposition 5.2 under $L_3$ Hessian-Lipschitz condition, with their proofs given in Appendix D.

**Proposition 5.1.** *For each $i$-th unlearning request $t_i \in U_t = \{t_1, \dots, t_k\}$, let $n_{t_i, s+1}^{t_i}$ denote the number of tasks in the time interval $[s + 1, t_i]$ that have been unlearned by time $t_i$. Alg. 2's unlearning model $w_t^{-\mathbf{S}_{1:t}}$ achieves the following approximation error upper bound $\gamma_t(\mathbf{S}_{1:t})$:*

$$\|w_t^{-S_{\leq t}} - w_t^{-\mathbf{S}_{1:t}}\| \leq \gamma_t(\mathbf{S}_{1:t}) :=$$

$$\frac{L(M-\mu)}{\lambda(\mu+\lambda)} \Bigg[ \sum_{s \in S_{t_k}} \rho^{t-s-n_{t_k,s}^k}(t_k - s - n_{t_k,s}^k) + \sum_{i=1}^{k-1} \sum_{s \in S_{t_i}} \rho^{t-s-n_{t_k,s}^k}(t_i - s - n_{t_i,s}^k) +$$

$$\sum_{i=1}^{k-1} \sum_{s \in S_{t_i}} \rho^{t-s-n_{t_k,s}^k}(1 - \rho^{n_{t_i,s}^k - n_{t_i,s}^i}) \sum_{x=0}^{k-i}(t_{k-x} - t_{k-x-1} - n_{t_{k-x}, t_{k-x-1}}^k) \left( \frac{M(\mu+\lambda)}{\lambda(M+\mu)} \right)^{k-i-x+1} \Bigg].$$

$$(14)$$

Proposition 5.1 shows that any unlearning request on a task $s$ introduces an error term proportional to $\rho^{t_i - s - n_{t_k,s}^k}$, similar in form to the unlearning loss in (9). For a newly unlearned task at time $t_i$, the unlearning procedure adds a term $\rho^{t-s-n_{t_k,s}^k}(t_i - s - n_{t_k,s}^k)\frac{L(M-\mu)}{\lambda(\mu+\lambda)}$. Then, for each task $s$ that was unlearned at earlier times prior to the most recent unlearning time $t_k$, with an example of $S_3 = \{2\}, S_5 = \{1, 3, 4\}$, it disrupts the unlearning sequence order, where we will have $\rho^{n_{t_i,s}^k - n_{t_i,s}^i} \neq 1$ in the last line of (14) and incurs the additional error term in the last line of (14). Otherwise, if unlearning requests arrive in a well-ordered sequence, e.g., $S_3 = \{2\}, S_5 = \{1, 3, 4\}$ such that

$\rho^{n^k_{t_i,s} - n^i_{t_i,s}} = 1$, the overall approximation error in (14) can be reduced. Hence, our Hessian-based algorithm is more sensitive to the unlearning sequence as compared to the natural forgetting algorithm in Alg. 1.

**Proposition 5.2.** *Suppose the loss function $\ell(w, z)$ is $L_3$ Hessian-Lipschitz. Alg. 2's unlearning model $w_t^{-\mathbf{S}_{1:t}}$ achieves the following second-order approximation error upper bound $\gamma_t(\mathbf{S}_{1:t})$:*

$$\|w_t^{-S_{\leq t}} - w_t^{-\mathbf{S}_{1:t}}\| \leq \gamma_t(\mathbf{S}_{1:t}) := \frac{L_3}{2} \sum_{i=\min S_{\leq t}+1, i\notin S_{\leq t}}^{t} \|w_m^{S_{\leq t}\setminus\{m+1,\dots,t\}} - w_m^{-\mathbf{S}_{1:m-1}}\|^2, \quad (15)$$

The key advantage of the Hessian-based algorithm in Alg. 2 over natural forgetting in Alg. 1 is its tighter second-order upper bound in Proposition 5.2. Note that the model approximation error in (15) can be decomposed into a natural forgetting error and the unlearning loss from previous unlearning times, which remain small under large $\lambda$ that restricts model updates. Consequently, the approximation error is typically below 1, allowing the second-order approximation to reduce it quadratically. Moreover, the unlearning loss vanishes when the loss is exactly quadratic, as in linear models with mean squared error.

Similar to Theorem 4.1, Alg. 2 guarantees the following performance.

**Corollary 5.3.** *Alg. 2 using noise coefficient $\gamma_t(\mathbf{S}_{1:t})$ in (14) or (15) guarantees $(\varepsilon, \delta)$-certified continual unlearning in Definition 2.1. Furthermore, the output model $\tilde{w}_t^{-\mathbf{S}_{1:t}}$ of Alg. 2 achieves the post-unlearning excess risk in (10) with $\mathcal{E}^{-S_{\leq t}}(\lambda)$ given in Theorem 3.1, and $\gamma_t(\mathbf{S}_{1:t})$ in (14) or (15).*

While Alg. 2 achieves a lower post-unlearning excess risk than Alg. 1, by time $t$ it incurs additional storage overhead of $O(td^2 + 2td)$ for storing the Hessian, model updates, and historical unlearning corrections. In contrast, Alg. 1 requires no storage. To reduce storage costs while keeping a relatively small post-unlearning excess risk, we next incorporate the natural forgetting effect utilized in Alg. 1 to improve Alg. 2.

### 5.3 Forgetting enhanced Hessian-based algorithm

We first derive the following results to motivate the combination methods in this section.

**Lemma 5.4.** *If the arrival of unlearning requests $U_t = \{t_1, \dots, t_k\}$, $S_{t_1}, \dots, S_{t_k}$ satisfy a certain retirement pattern that all the unlearning tasks in $S_{t_i}$ requested at each time $t_i$ arrive after the last unlearning request's time $t_{i-1}$, i.e., $t_{i-1} \leq s \leq t_i$, $\forall s \in S_{t_i}$, $t_{i-1}, t_i \in U_t$, the unlearning correction in (13) simplies to $\bar{\Delta}_t = \sum_{s\in S_t}(\prod_{i=s+1, i\notin S_{\leq t}}^{t}(H_i + \lambda I)^{-1}\lambda I)\Delta_s$.*

When unlearning requests arrive in this well-ordered manner, each new request does not interrupt the previous continual learning order, where we do not need to use the second line in (13) to adjust for the influence of new unlearning requests, and thus no information from tasks trained before the last unlearning time needs to be stored. The proof of Lemma 5.4 given in Appendix D.3

Although unlearning sequences typically do not arrive in such an ordered fashion, we can still combine our two methods: using the Hessian-based approach to unlearn requests involving tasks trained after the last unlearning time, and relying on natural forgetting in continual learning to handle requests involving tasks trained earlier. To achieve this, we introduce the following modification to line 10 and (13) in Alg. 2.

- Denote $S_t' = \{s \in S_t : s \geq \max U_{t-1}\}$, update

$$w_t^{-\mathbf{S}_{1:t}} = w_t^{-\mathbf{S}_{1:t-1}} + \sum_{s\in S_t'}(\prod_{i=s+1, i\notin S_{\leq t}}^{t}(H_i + \lambda I)^{-1}\lambda I)\Delta_s.$$

  Then, discard all the stored information on tasks before $t$.

Under this modification to Alg. 2, the storage overhead reduces to $\max_{t_i, t_{i-1}\in U_t}(t_i - t_{i-1})(d^2 + 2d)$, proportional to the maximum distance between any two concecutive unlearning time. We show the performance guarantee of this modified Alg. 2 in Appendix D.3.

## 6 EXPERIMENTS

We perform experiments on the standard MNIST dataset. To generate the dataset for continual learning, we split the 60,000-sample training set into $T = 30$ non-i.i.d. sequential tasks, where each task contains data from at most three labels randomly picked out of the ten digits $0-9$. We train a linear model with a softmax output under the cross-entropy loss. Regarding Assumption 2.1, we relax its assumption of $\mu$-strong convexity here in order to show the more general results under a non–strongly convex setting.

### 6.1 BALANCING BETWEEN EXCESS RISK AND UNLEARNING LOSS

We first run the continual learning algorithm (Alg. 1) without unlearning to tell excess risk and then apply our unlearning algorithms (Alg. 1 and 2) to examine unlearning loss under a randomly generated unlearning sequence (the first row of Table 2) under different values of $\lambda$. Fig. 2 (a) shows the digit recognition accuracy on the test set, which represents the excess risk in (7); Fig. 2 (b) shows the approximation error under two algorithms, which represents the unlearning loss in (6).

As in the upper bounds in Theorems 4.1 and Corollary 5.3 for the two unlearning algorithms, we need to choose $\lambda$ to balance excess risk and unlearning loss. As shown in Fig. 2(a), the optimal $\lambda$ for excess risk minimization lies in the range $[5, 10]$, in contrast to $\lambda = 40$ and $\lambda = 20$, which minimize the approximation error for unlearning under the natural forgetting and Hessian-based algorithms, respectively.

To evaluate the post-unlearning excess risk, we measure the test accuracy of $\tilde{w}_T^{-\mathbf{S}_{1:T}}$ produced by Alg. 2 under the first unlearning sequence, as well as that of the retrained model, which serves as the loose accuracy upper bound. Our Hessian-based algorithm reaches its maximum test accuracy of 71.59% at $\lambda = 30$, since it does not rely on forgetting to facilitate unlearning.

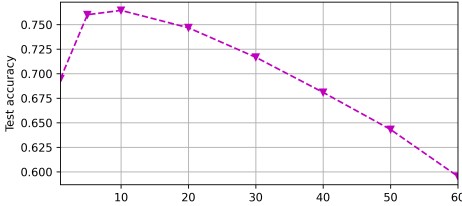 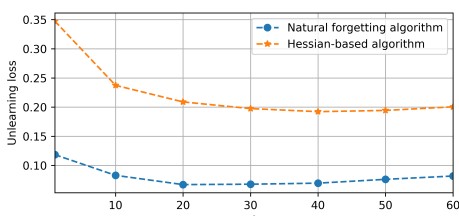

Figure 2: (a) Test accuracy for excess risk examination after training on all 30 tasks using algorithm (1) without unlearning. (b) Approximation error for unlearning loss examination under the unlearning sequence in the first row of Table 2 returned by Alg. 1 and Alg. 2.

|  | Hessian-based unlearning | Perfect retraining |
|---|---|---|
| $\lambda = 10$ | 64.17% | 74.11% |
| $\lambda = 20$ | 67.59% | 75.81% |
| $\lambda = 30$ | 71.59% | 71.05% |

Table 1: Test accuracy for evaluating post-unlearning excess risk.

We also present experiments in Appendix E to demonstrate the influence of unlearning sequence patterns to the unlearning algorithms.

## 7 CONCLUSION

We established the first theoretical foundation connecting continual learning and machine unlearning, reconciling the need to preserve past knowledge with efficient forgetting under unlearning requests. By adapting gradient- and Hessian-based certified unlearning methods to the continual learning setting, we derived theoretical guarantees on both excess risk and unlearning loss, which jointly determine post-unlearning performance. Our analysis shows that the Hessian-based method achieves lower unlearning loss, while the gradient-based method eliminates storage costs. Experiments on MNIST validate our theory and highlight the impact of unlearning sequence patterns.

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

## ETHICS STATEMENT

This work does not involve human subjects, sensitive personal data, or any potentially harmful applications. The datasets used are publicly available benchmark datasets (e.g., MNIST), and all experiments strictly follow standard machine learning practices. There are no foreseeable risks regarding fairness, bias, privacy, security, or legal compliance.

## REPRODUCIBILITY STATEMENT

We provide full details of our theoretical results, including all assumptions and complete proofs, in the appendix.

## LLM USAGE STATEMENT

Large Language Models (LLMs) were used only for language polishing and improving the clarity of writing. No LLMs were used in research ideation, algorithm design, experimental analysis, or result generation. The authors take full responsibility for the content of this paper.

## A  RELATED WORKS

In continual learning, an increasing number of recent works have begun to investigate the theoretical performance of forgetting and generalization loss induced by continual learning algorithms (Evron et al. (2022); Lin et al. (2023); Swartworth et al. (2023); Zhao et al. (2024); Deng et al. (2025)). Due to the inherent difficulty of the continual learning problem, most prior work has focused on simple regularization-based continual learning algorithms and linear models for the ease of deriving theoretical results.

Current studies that examine continual learning and unlearning together are experimental, lacking theoretical guarantees (Liu et al. (2022); Chatterjee et al. (2024); Cha et al. (2024); Huang et al. (2025)). For example, Chatterjee et al. (2024) uses a controlled knowledge distillation where a continual learning teacher preserves past knowledge while another unlearning teacher drives targeted forgetting, both guiding a student model to update. Huang et al. (2025) proposes a gradient descent

approach that achieves both unlearning and continual learning by designing a combined loss function, while assuming that the dataset to be unlearned is available again at the time of the unlearning request. These methods merely forget the designated tasks, whereas certified continual unlearning demands a strict approximation of the model that would result from perfect retraining. This highlights the need to develop and rigorously analyze unlearning algorithms for continual learning that can remove data influence without requiring access to the original dataset.

Basaran et al. (2025) study certified unlearning without access to the original dataset by using a surrogate dataset. However, their work is limited to the static learning case, and their method cannot be adapted to continual learning, where the model evolves with more complex dynamics.

# B  MISSING DETAILS FROM SECTION 3

## B.1  PROOF OF THEOREM 3.1

Without loss of generality, we prove the results on the sequence $\{1, \ldots, t\}$, which can be generalized to any retraining sequence $\{\tau_1, \ldots, \tau_k\} = \{1, \ldots, t\} \setminus S_{\leq t}$. We first present two preliminary results in Lemmas B.1 and B.2, and then prove Theorem 3.1 using these lemmas.

**Lemma B.1.** *Let* $w_t^* = \arg\min_w F_t(w) = \arg\min_w \mathbb{E}_{z \sim \mathcal{D}_t}[\ell(w, z)]$ *denote the population-optimal model for task* $t$, *and* $\hat{w}_t = \arg\min_w \tilde{F}_t(w) = \arg\min_w \sum_{i \in D_t} \ell(w, z_i)$ *denote the corresponding empirical risk minimizer. Then we have*

$$\mathbb{E}[\|w_t^* - \hat{w}_t\|] \leq \frac{2\sqrt{2}L}{\mu\sqrt{|D_t|}}.$$

*Proof.* From Shalev-Shwartz et al. (2009), it follows that

$$\mathbb{E}\big[F_t(\hat{w}_t) - F_t(w_t^*)\big] \leq \frac{4L^2}{\mu|D_t|}.$$

Moreover, since $F_t$ is $\mu$-strongly convex, we have

$$F_t(\hat{w}_t) - F_t(w_t^*) \geq \frac{\mu}{2}\|\hat{w}_t - w_t^*\|^2.$$

Taking expectations on both sides gives

$$\frac{\mu}{2}\mathbb{E}\big[\|\hat{w}_t - w_t^*\|\big]^2 \leq \frac{\mu}{2}\mathbb{E}\big[\|\hat{w}_t - w_t^*\|^2\big] \leq \mathbb{E}\big[F_t(\hat{w}_t) - F_t(w_t^*)\big] \leq \frac{4L^2}{\mu|D_t|}.$$

Rearranging yields the desired bound. $\qquad\square$

**Lemma B.2.** *Let* $w_t$ *denote the model obtained by running (1) on tasks* $D_{1:t}$ *in the continual learning setting. Let* $w_t^* = \arg\min_w F_t(w) = \arg\min_w \mathbb{E}_{z \sim \mathcal{D}_t}[\ell(w, z)]$ *denote the population-optimal model for task* $t$, *and* $\hat{w}_t = \arg\min_w \tilde{F}_t(w) = \arg\min_w \sum_{i \in D_t} \ell(w, z_i)$ *denote the corresponding empirical risk minimizer. Then we have the following for any* $m > 1$:

$$\mathbb{E}[\|w_t - \hat{w}_m\|] \leq \sum_{i=2, i\neq m}^{t} \frac{M}{M+\lambda}\left(\frac{\lambda}{\lambda+\mu}\right)^{t-i}\left(\|w_i^* - w_m^*\| + \frac{2\sqrt{2}L}{\mu}\left(\frac{1}{|D_i|} + \frac{1}{|D_m|}\right)\right)$$

$$+ \left(\frac{\lambda}{\lambda+\mu}\right)^{t}\left(\|w_1^* - w_m^*\| + \frac{2\sqrt{2}L}{\mu}\left(\frac{1}{|D_1|} + \frac{1}{|D_m|}\right)\right)$$

$$+ \frac{M}{M+\lambda}\left(\frac{\lambda}{\lambda+\mu}\right)^{t-1}\left(\|w_m^*\| + \frac{2\sqrt{2}L}{\mu}\frac{1}{|D_m|}\right),$$

*and*

$$\mathbb{E}[\|w_t - \hat{w}_1\|] \leq \sum_{i=2}^{t} \frac{M}{M+\lambda}\left(\frac{\lambda}{\lambda+\mu}\right)^{t-i}\left(\|w_i^* - w_1^*\| + \frac{2\sqrt{2}L}{\mu}\left(\frac{1}{|D_i|} + \frac{1}{|D_1|}\right)\right)$$

$$+ \frac{M}{M+\lambda}\left(\frac{\lambda}{\lambda+\mu}\right)^{t-1}\left(\|w_1^*\| + \frac{2\sqrt{2}L}{\mu}\frac{1}{|D_1|}\right).$$

*Proof.* According to the training algorithm in Alg. 1, we have

$$\nabla \tilde{F}_t(w_t) + \lambda(w_t - w_{t-1}) = 0.$$

Letting $\nabla \tilde{F}_t(w_t)$ Taylor expand at the point of $\hat{w}_t$, then

$$\nabla \tilde{F}_t(\hat{w}_t) + \int_0^1 \nabla^2 \tilde{F}_t(\hat{w}_t + u(w_t - \hat{w}_t))du(w_t - \hat{w}_t) + \lambda(w_t - w_{t-1}) = 0$$

$$w_t = \Big(\int_0^1 \nabla^2 \tilde{F}_t(\hat{w}_t + u(w_t - \hat{w}_t))du + \lambda I\Big)^{-1}\Big(\int_0^1 \nabla^2 \tilde{F}_t(\hat{w}_t + u(w_t - \hat{w}_t))du\ \hat{w}_t + \lambda w_{t-1}\Big).$$

Denote $\tilde{H}_t = \int_0^1 \nabla^2 \tilde{F}_t(\hat{w}_t + u(w_t - \hat{w}_t))du$. According to the iteration above, we can write $w_t$ as the following:

$$w_t = \sum_{i=2}^t \Big(\prod_{j=i+1}^t (\tilde{H}_j + \lambda I)^{-1}\lambda I\Big)(\tilde{H}_i + \lambda I)^{-1}\tilde{H}_i \hat{w}_i + \Big(\prod_{j=1}^t (\tilde{H}_j + \lambda I)^{-1}\lambda I\Big)\hat{w}_1,$$

where the second term arises because $w_1$ is the first updated model obtained without the regularization term in (1). Then, we can upper bound $\mathbb{E}[\|w_t - \hat{w}_m\|]$ for any for $m > 1$ as follows:

$$\mathbb{E}[\|w_t - \hat{w}_m\|]$$

$$= \mathbb{E}\Big[\Big\| \sum_{i=2,i\neq m}^t \Big(\prod_{j=i+1}^t (\tilde{H}_j + \lambda I)^{-1}\lambda I\Big)(\tilde{H}_i + \lambda I)^{-1}\tilde{H}_i(\hat{w}_i - \hat{w}_m)$$

$$+ \Big(\prod_{j=1}^t (\tilde{H}_j + \lambda I)^{-1}\lambda I\Big)(\hat{w}_1 - \hat{w}_m) - \Big(\prod_{j=1}^t (\tilde{H}_j + \lambda I)^{-1}\lambda I\Big)(\tilde{H}_1 + \lambda I)^{-1}\tilde{H}_1 \hat{w}_m\Big\|\Big]$$

$$\leq \mathbb{E}\Big[ \sum_{i=2,i\neq m}^t \Big\|\Big(\prod_{j=i+1}^t (\tilde{H}_j + \lambda I)^{-1}\lambda I\Big)(\tilde{H}_i + \lambda I)^{-1}\tilde{H}_i(\hat{w}_i - \hat{w}_m)\Big\|$$

$$+ \Big\|\Big(\prod_{j=1}^t (\tilde{H}_j + \lambda I)^{-1}\lambda I\Big)(\hat{w}_1 - \hat{w}_m)\Big\| + \Big\|\Big(\prod_{j=w}^t (\tilde{H}_j + \lambda I)^{-1}\lambda I\Big)(\tilde{H}_1 + \lambda I)^{-1}\tilde{H}_1 \hat{w}_m\Big\|\Big]$$

$$= \sum_{i=2,i\neq m}^t \frac{M}{M+\lambda}\Big(\frac{\lambda}{\lambda+\mu}\Big)^{t-i} \mathbb{E}[\|\hat{w}_i - \hat{w}_m\|] + \Big(\frac{\lambda}{\lambda+\mu}\Big)^t \mathbb{E}[\|\hat{w}_1 - \hat{w}_m\|]$$

$$+ \frac{M}{M+\lambda}\Big(\frac{\lambda}{\lambda+\mu}\Big)^{t-1} \mathbb{E}[\|\hat{w}_m\|]$$

$$\leq \sum_{i=2,i\neq m}^t \frac{M}{M+\lambda}\Big(\frac{\lambda}{\lambda+\mu}\Big)^{t-i} \mathbb{E}[\|\hat{w}_i - w_i^*\| + \|\hat{w}_m - w_m^*\| + \|w_i^* - w_m^*\|]$$

$$+ \Big(\frac{\lambda}{\lambda+\mu}\Big)^t \mathbb{E}[\|\hat{w}_1 - w_1^*\| + \|\hat{w}_m - w_m^*\| + \|w_1^* - w_m^*\|]$$

$$+ \frac{M}{M+\lambda}\Big(\frac{\lambda}{\lambda+\mu}\Big)^{t-1} \mathbb{E}[\|\hat{w}_m - w_m^*\| + \|w_m^*\|],$$

where the last upper bound is obtained by $\|(\tilde{H}_j + \lambda I)^{-1}\tilde{H}_j\| \leq M/(M+\lambda)$ and $\|(\tilde{H}_j + \lambda I)^{-1}\lambda\| \leq \lambda/(\mu + \lambda)$ from Assmption 2.1. By applying Lemma B.1, we obtain the result for $m > 1$; the case $m = 1$ follows by an analogous proof. $\qquad\square$

We proceed to prove Theorem 3.1. To begin, we have

$$\mathbb{E}[\sum_{\tau=1}^{t} F_\tau(w_t) - \min_w \sum_{\tau=1}^{t} F_\tau(w)] \leq \mathbb{E}[\sum_{\tau=1}^{t} F_\tau(w_t) - \sum_{\tau=1}^{t} \min_w F_\tau(w)]$$

$$= \mathbb{E}[\sum_{\tau=1}^{t} F_\tau(w_t) - F_\tau(w_i^*)]$$

$$= \mathbb{E}[\sum_{\tau=1}^{t} F_\tau(w_t) - F_\tau(\hat{w}_\tau) + F_\tau(\hat{w}_\tau) - F_\tau(w_i^*)]$$

$$\leq E[\sum_{\tau=1}^{t} L\|w_t - \hat{w}_\tau\| + \frac{4L^2}{\mu|D_t|}],$$

where the last line follows from the $L$-Lipschitz property of $F_t$ and Lemma B.1. Substituting Lemma B.2 above and rearranging yields the desired result.

## B.2 EXAMPLES OF THEOREM 3.1

For simplicity, assume $n_t = |D_t|$ is sufficiently large for all $t$. Consider $k = 3$ with $M = \mu$, where (8) simplifies to

$$L[\|w_{\tau_1}^* - w_{\tau_3}^*\| + \|w_{\tau_2}^* - w_{\tau_3}^*\| + \rho^3(\|w_{\tau_1}^* - w_{\tau_2}^*\| + \|w_{\tau_1}^* - w_{\tau_3}^*\| - \|w_{\tau_1}^*\| - \|w_{\tau_2}^*\| - \|w_{\tau_3}^*\|)$$

$$+ \rho^2(\|w_{\tau_1}^*\| + \|w_{\tau_2}^*\| + \|w_{\tau_3}^*\| - \|w_{\tau_1}^* - w_2^*\|) + \rho(\|w_{\tau_{\tau_1}}^* - w_{\tau_2}^*\| - \|w_{\tau_1}^* - w_{\tau_3}^*\| - \|w_{\tau_2}^* - w_{\tau_3}^*\|)].$$

If $\|w_{\tau_1}^* - w_{\tau_2}^*\| + \|w_{\tau_1}^* - w_{\tau_3}^*\| - \|w_{\tau_1}^*\| - \|w_{\tau_2}^*\| - \|w_{\tau_3}^*\| = 0$ and $\|w_{\tau_1}^* - w_{\tau_2}^*\| - \|w_{\tau_1}^* - w_{\tau_3}^*\| - \|w_{\tau_2}^* - w_{\tau_3}^*\| < 0$, then we require $0 < \lambda$ to minimize excess risk.

Another case to consider is the following. when $\rho = 0$ ($\lambda = 0$), the bound reduces to $\sum_{i=1}^{k} \|w_{\tau_i}^* - w_{\tau_k}^*\|$; when $\rho = 1$ ($\lambda \to \infty$), it becomes $\sum_{i=1}^{k} \|w_{\tau_i}^* - w_{\tau_1}^*\|$. Thus, if the first task differs more from later tasks than the last task, the excess risk is must be minimized for some $\lambda > 0$. This is consistent with the findings in Lin et al. (2023) under the linear continual learning model.

## C MISSING DETAILS FROM SECTION 4

### C.1 PROOF OF THEOREM 4.1

We first prove the approximation error $\|w_t - w_t^{-S_{\leq t}}\|$'s upper bound for $\gamma_t(\mathbf{S}_{1:t})$ in (9). Write $S_{\leq t} = \{s_1, \ldots, s_n\}$, where the elements are ordered increasingly. According to continual learning algorithm (1), we have the following first-order conditions:

$$\nabla \tilde{F}_t(w_t^{-S_{\leq t}}) + \lambda(w_t^{-S_{\leq t}} - w_{t-1}^{-S_{\leq t}}) = 0, \ \nabla \tilde{F}_t(w_t) + \lambda(w_t - w_{t-1}) = 0.$$

Having $\nabla \tilde{F}_t(w_t^{-S_{\leq t}})$ Taylor expansion at the point of $w_t$, we have

$$\nabla \tilde{F}_t(w_t) + \int_0^1 \nabla^2 \tilde{F}_t(w_t + u(w_t^{-S_{\leq t}} - w_t))du(w_t^{-S_{\leq t}} - w_t) + \lambda(w_t^{-S_{\leq t}} - w_{t-1}^{-S_{\leq t}}) = 0$$

$$- \lambda(w_t - w_{t-1}) + \int_0^1 \nabla^2 \tilde{F}_t(w_t + u(w_t^{-S_{\leq t}} - w_t))du(w_t^{-S_{\leq t}} - w_t) + \lambda(w_t^{-S_{\leq t}} - w_{t-1}^{-S_{\leq t}}) = 0$$

$$w_t^{-S_{\leq t}} - w_t = (\int_0^1 \nabla^2 \tilde{F}_t(w_t + u(w_t^{-S_{\leq t}} - w_t))du + \lambda I)^{-1}\lambda(w_{t-1}^{-S_{\leq t}} - w_{t-1})$$

Let $\tilde{H}_t = \int_0^1 \nabla^2 \tilde{F}_t\left(w_t + u(w_t^{-S_{\leq t}} - w_t)\right) du$. Keep doing the iteration above, we have

$$w_t^{-S_{\leq t}} - w_t = (\prod_{i=s_n+1}^{t} (\tilde{H}_t + \lambda I)^{-1}\lambda I)(w_{s_n}^{-S_{\leq t}} - w_{s_n})$$

Note that $w_{s_n}^{-S_{\leq t}} = w_{s_n-1}^{-S_{\leq t}\backslash\{s_n\}}$ since task $s_n$ is removed when retraining on all remaining tasks. Then,

$$w_t^{-S_{\leq t}} - w_t = (\prod_{i=s_n+1}^{t} (\tilde{H}_t + \lambda I)^{-1}\lambda I)(w_{s_n}^{-S_{\leq t}} - w_{s_n})$$

$$= (\prod_{i=s_n+1}^{t} (\tilde{H}_t + \lambda I)^{-1}\lambda I)(w_{s_n-1}^{-S_{\leq t}\backslash\{s_n\}} - w_{s_n-1} + w_{s_n-1} - w_{s_n})$$

$$= (\prod_{i=s_n+1}^{t} (\tilde{H}_t + \lambda I)^{-1}\lambda I)(w_{s_n-1}^{-S_{\leq t}\backslash\{s_n\}} - w_{s_n-1}) + \prod_{i=s_n+1}^{t} (w_{s_n-1} - w_{s_n})$$

Keep doing this iteration for all tasks $s_i \in S_t$, it becomes

$$w_t^{-S_{\leq t}} - w_t$$

$$= (\prod_{i=s_n+1}^{t} (\tilde{H}_i + \lambda I)^{-1}\lambda I)(w_{s_n}^{-S_{\leq t}} - w_{s_n})$$

$$= (\prod_{i=s_n+1}^{t} (\tilde{H}_i + \lambda I)^{-1}\lambda I)(w_{s_n-1}^{-S_{\leq t}\backslash\{s_n\}} - w_{s_n-1} + w_{s_n-1} - w_{s_n})$$

$$= (\prod_{i=s_1+1, i\notin S_{\leq t}}^{t} (\tilde{H}_i + \lambda I)^{-1}\lambda I)(w_{s_1-1}^{-S_{\leq t}\backslash\{s_1,\dots,s_n\}} - w_{s_1-1})$$

$$+ \sum_{j=1}^{n} (\prod_{i=s_j+1, i\notin S_{\leq t}}^{t} (\tilde{H}_i + \lambda I)^{-1}\lambda I)(w_{s_j-1} - w_{s_j})$$

$$= (\prod_{i=s_1+1, i\notin S_{\leq t}}^{t} (\tilde{H}_i + \lambda I)^{-1}\lambda I)(w_{s_1-1}^{-\emptyset} - w_{s_1-1}) + \sum_{j=1}^{n} (\prod_{i=s_j+1, i\notin S_{\leq t}}^{t} (\tilde{H}_i + \lambda I)^{-1}\lambda I)(w_{s_j-1} - w_{s_j})$$

$$= \sum_{j=1}^{n} (\prod_{i=s_j+1, i\notin S_{\leq t}}^{t} (\tilde{H}_i + \lambda I)^{-1}\lambda I)(w_{s_j-1} - w_{s_j})$$

Taking the norm of both side, we have

$$\|w_t^{-S_{\leq t}} - w_t\| = \|\sum_{j=1}^{n} (\prod_{i=s_j+1, i\notin S_{\leq t}}^{t} (\tilde{H}_i + \lambda I)^{-1}\lambda I)(w_{s_j-1} - w_{s_j})\|$$

$$\leq \sum_{j=1}^{n} \prod_{i=s_j+1, i\notin S_{\leq t}}^{t} \|(\tilde{H}_i + \lambda I)^{-1}\lambda I\|\|(w_{s_j-1} - w_{s_j})\| \qquad (16)$$

For each $\|(\tilde{H}_i + \lambda I)^{-1}\lambda I\|$, by the $\mu$-strong convex assumption, we have $\|(\tilde{H}_i + \lambda I)^{-1}\lambda I\| \leq \lambda/(\mu+\lambda)$ for any $i$. For the $\|(w_{s_j-1} - w_{s_j})\|$, according to the first order condition under algorithm 1, we have

$$w_{s_j-1} - w_{s_j} = \frac{-\nabla F_{s_j}(w_{s_j})}{\lambda} \implies \|w_{s_j-1} - w_{s_j}\| \leq \frac{L}{\lambda},$$

where we use $\|\nabla F_{s_j}(w_{s_j})\| < L$ which follows from the $L$-Lipschitz condition. By substituting the upper bounds for $\|(\tilde{H}_i + \lambda I)^{-1}\lambda I\|$ and $\|\nabla F_{s_j}(w_{s_j})\|$ into (16), we obtain the result in (9).

Given the upper bound of the approximation error, the noise mechanism in line 12 of Alg. 1 ensures $(\varepsilon, \delta)$-certified unlearning, following the standard proofs in certified unlearning (e.g., Dwork et al. (2014); Qiao et al. (2024)).

Then, to establish the post-unlearning excess risk, it suffices, by the decomposition in (7) and (6), to prove an upper bound for the unlearning loss in the first term of (6):

$$
\mathbb{E}\left[\frac{1}{t - |S_{\leq t}|} \sum_{\tau=1, \tau \notin S_{\leq t}}^{t} F_{\tau}(\tilde{w}_t^{-\mathbf{S}_{1:t}}) - F_{\tau}(w_t^{-S_{\leq t}})\right]
$$

$$
\leq \mathbb{E}\left[\frac{1}{t - |S_{\leq t}|} \sum_{\tau=1, \tau \notin S_{\leq t}}^{t} L\|\tilde{w}_t^{-\mathbf{S}_{1:t}} - w_t^{-S_{\leq t}}\|\right]
$$

$$
= \mathbb{E}\left[L\|\tilde{w}_t^{-\mathbf{S}_{1:t}} - w_t^{-S_{\leq t}}\|\right]
$$

$$
\leq \mathbb{E}\left[L\|\tilde{w}_t^{-\mathbf{S}_{1:t}} - w_t^{-\mathbf{S}_{1:t}}\|\right] + \mathbb{E}\left[L\|w_t^{-\mathbf{S}_{1:t}} - w_t^{-S_{\leq t}}\|\right]
$$

$$
= L\sqrt{\mathbb{E}\left[\|\tilde{w}_t^{-\mathbf{S}_{1:t}} - w_t^{-\mathbf{S}_{1:t}}\|\right]^2} + L\mathbb{E}\left[\|w_t^{-\mathbf{S}_{1:t}} - w_t^{-S_{\leq t}}\|\right]
$$

$$
\leq L\sqrt{\mathbb{E}\left[\|\tilde{w}_t^{-\mathbf{S}_{1:t}} - w_t^{-\mathbf{S}_{1:t}}\|^2\right]} + L\gamma_t(\mathbf{S}_{1:t})
$$

$$
= L\gamma_t(\mathbf{S}_{1:t})\sqrt{2d\ln(1.25/\delta)}/\varepsilon + L\gamma_t(\mathbf{S}_{1:t}),
$$

where the first inequality follows from the $L$-Lipschitz condition, the last inequality from Jensen's inequality, and the final step from substituting the variance defined in line 12 of Alg. 1. Then we complete the proof.

## C.2 A STRONGER VERSION OF $(\varepsilon, \delta)$-CERTIFIED UNLEARNING

To further unlearn the model and ensure certified unlearning even at the system side, we extend Alg. 1 by removing all internal information of deleted tasks between iterations. Specifically, we make two changes to Alg. 1:

- After computing $\tilde{w}_t^{-\mathbf{S}_{1:t}} = f(w_t^{-\mathbf{S}_{1:t}})$ in line 12, we discard $w_t$ and use $\tilde{w}_t^{-\mathbf{S}_{1:t}}$ instead for training on next task $t + 1$.
- After training on task $t + 1$ starting from noisy $\tilde{w}_t^{-\mathbf{S}_{1:t}}$, the updated model $w_{t+1}^{-\mathbf{S}_{1:t}}$ still satisfies the certified unlearning guarantee relative to the perfectly retraining model $w_{t+1}^{-S_{\leq t}}$, by the post-processing immunity of differential privacy Dwork et al. (2014). Therefore, we only need to inject noise when a nonempty deletion request $S_t$ arrives, changing the condition in line 8 from $S_{\leq t} = \emptyset$ to $S_t = \emptyset$.

With these modifications, Theorem 4.1 extends to the corollary below for strong privacy protection.

**Corollary C.1.** *For each $i$-th unlearning request $t_i \in U_t = \{t_1, \ldots, t_k\}$, let $n_{t_i, s+1}^{t_i}$ denote the number of tasks in the interval $[s + 1, t_i]$ that have been deleted by time $t$. Then,*

(1) *With probability at least $1 - \delta$, the modified Alg. 1's approximation error $\|\tilde{w}_t^{-\mathbf{S}_{1:t}} - w_t^{-S_{\leq t}}\|$ is upper bounded by $\gamma_t(\mathbf{S}_{1:t})$ below:*

$$
\gamma_t(\mathbf{S}_{1:t}) := \sum_{i=1}^{k} \sum_{s \in S_{t_i}} \rho^{t-s-n_{t,s+1}^k} \frac{L}{\lambda}(1 + \xi)^{k-i}, \tag{17}
$$

*where $\xi = \dfrac{d\sqrt{2\ln(1.25/\delta)\ln(dT/\delta)}}{\varepsilon}$.*

(2) *The modified Alg. 1 that operates with the $\gamma_t(\mathbf{S}_{1:t})$ guarantees $(\varepsilon, \delta)$-certified continual unlearning in Definition 2.1 with repect to both the public model $\tilde{w}_t^{-\mathbf{S}_{1:t}}$ and the internal state of the system, with probability at least $1 - \delta$.*

(3) *The output model $\tilde{w}_t^{-\mathbf{S}_{1:t}}$ of modified Alg. 1 achieves a post-unlearning excess risk in (5) that is upper bounded by the following expression with probability at least $1 - \delta$:*

$$
L\left(\frac{\sqrt{2d\ln(\frac{1.25}{\delta})}}{\varepsilon} + 1\right)\gamma_t(\mathbf{S}_{1:t}) + \mathcal{E}^{-S_{\leq t}}(\lambda),
$$

*where $\mathcal{E}^{-S_{\leq t}}(\lambda)$ is given in Theorem 3.1, and $\gamma_t(\mathbf{S}_{1:t})$ is in (17).*

$\gamma_t(\mathbf{S}_{1:t})$ in (17) is a high-probability upper bound that carries an extra multiplicative factor $(1 + \xi)^{k-i}$ relative to (9). This arises because, in our stronger privacy protection, we feed the noised model $\tilde{w}_t^{-\mathbf{S}_{1:t}}$ into the next task, which, in turn, increases the required noise to mask the unlearning loss at next unlearning time, so each round's injected noise accumulates and increases over the learning–unlearning cycles, adding extra random deviation.

## D  MISSING DETAILS FROM SECTION 5

### D.1  PROOF OF PROPOSITION 5.1

We use the following lemma, which provides a tight upper bound of the approximation error from Alg. 2 that depends sensitively on the unlearning sequence $\mathbf{S}_{1:t}$, to prove Proposition 5.1.

**Lemma D.1.** *For each $i$-th unlearning request $t_i \in U_t = \{t_1, \ldots, t_k\}$, let $n^i_{t_i,\,s+1}$ denote the number of tasks in the interval $[s+1,\,t_i]$ that have been deleted by time $t_i$. Alg. 2's approximation error $\|w_t^{-\mathbf{S}_{1:t}} - w_t^{-S_{\leq t}}\|$ is upper bounded by $\gamma_t(\mathbf{S}_{1:t})$ below:*

$$\gamma_t(\mathbf{S}_{1:t}) := \Bigg[ \sum_{s \in S_{t_k}} \rho^{t_k - s - n^k_{t_k,s}}(t_k - s - n^k_{t_k,s}) + \sum_{i=1}^{k-1} \sum_{s \in S_{t_i}} \rho^{t_k - s - n^k_{t_k,s}} \Bigg( (t_i - s - n^k_{t_i,s})$$

$$+ \sum_{x=0}^{k-i} (t_{k-x} - t_{k-x-1} - n^k_{t_{k-x},t_{k-x-1}}) C^x_{k-i}(t_k, t_{t_i}, s) \Bigg) \Bigg] \frac{L(M-\mu)}{\lambda(\mu+\lambda)}, \quad (18)$$

*where $C^x_{k-i}(t_k, t_{t_i}, s)$ follows the following iteration:*

$$C^x_{k-i}(t_k, t_{t_i}, s)$$

$$= \begin{cases} C^x_{k-i-1}(t_k, t_{i+1}, t_i) \frac{M(\mu+\lambda)}{\lambda(M+\mu)} \rho^{n^k_{t_i,s} - n^{i+1}_{t_i,s}}(1 - \rho^{n^{i+1}_{t_i,s} - n^i_{t_i,s}}) + C^x_{k-i-1}(t_k, t_{t_i}, s), \\ \quad \text{if } x = k-i+1, \\ C^{x+1}_{k-i-1}(t_k, t_{i+1}, t_i) \frac{M(\mu+\lambda)}{\lambda(M+\mu)} \rho^{n^k_{t_i,s} - n^{i+1}_{t_i,s}}(1 - \rho^{n^{i+1}_{t_i,s} - n^i_{t_i,s}}) + C^x_{k-i-1}(t_k, t_{t_i}, s), \\ \quad \text{if } x = 1, \ldots, k-i, \end{cases}$$

$$C^1_1(t_k, y, s) = \frac{M(\mu+\lambda)}{\mu(M+\lambda)}(1 - \rho^{n^k_{x,s} - n^{k-1}_{x,s}}) \text{ and } C^2_1(t_k, y, s) = 1. \quad (19)$$

Lemma D.1 shows that, for a task unlearned at time $t_i$, if at a later time $t_{i+1}$ the system receives a new unlearning request for a task between $s$ and $t_{i+1}$ (e.g., $n^{i+1}_{t_i,s} = n^i_{t_i,s}$), an additional error term is incurred of the form $C^{k-i}_{k-i}(t_k, t_i, s)\,(t_{i+1} - t_i - n^k_{t_{i+1},t_i})$. If further unlearning requests arrive between $t_i$ and $s$, the parameter $C^{k-i}_{k-i}(t_k, t_i, s)$ will continue to grow according to the recurrence of $C$ defined in (19). Also, further unlearning interaption between later time $t_{j+1}$ and $t_j$ will continue to add terms of $t_{j+1} - t_j - n^k_{t_{j+1},t_j}$ for task $s$.

*proof of Proposition 5.1.* To complete the proof of Proposition 5.1, it remains to prove the following inequality:

$$C^x_{k-i}(t_k, t_{t_i}, s) \leq \rho^{t - s - n^k_{t_k,s}}(1 - \rho^{n^k_{t_i,s} - n^i_{t_i,s}}) \left( \frac{M(\mu+\lambda)}{\lambda(M+\mu)} \right)^{k-i-x+1}$$

for any $x = 1, \ldots, k-i+1$ and $i = 1, \ldots, k-1$. We present the proof for $C^1_m$ as an example, where the same argument applies to all other $C^t_{k-i}$.

$$C^1_m(t_k, t_{k-m}, s) \leq \rho^{t_k - s - n^k_{t_k,s}}(1 - \rho^{n^k_{t_{k-m},s+1} - n^{k-m}_{t_{k-m},s+1}}) \left( \frac{M(\mu+\lambda)}{\mu(M+\lambda)} \right)^m.$$

For $m = 1$, the claim follows directly from the definition of $C_1^1(t_k, y, s)$ in (19). Suppose the claim holds for some $m$. Then, for $m + 1$, by the iteration in (19):

$$C_{m+1}^1(t_k, t_{k-m-1}, s)$$

$$\leq C_m^1(t_k, t_{k-m-1}, s)$$

$$+ C_m^1(t_k, t_{k-m}, t_{k-m-1}) \frac{M(\mu + \lambda)}{\mu(M + \lambda)} \rho^{t_{k-m-1} - s - n_{t_{k-m-1}, s}^{k-m}} \left(1 - \rho^{n_{t_{k-m-1}, s+1}^{k-m} - n_{t_{k-m-1}, s+1}^{k-m}}\right)$$

$$= \rho^{t_k - s - n_{t_k, s}^k} \left(\frac{M(\mu + \lambda)}{\mu(M + \lambda)}\right)^m \left(1 - \rho^{n_{t_{k-m-1}, s+1}^k - n_{t_{k-m-1}, s+1}^{k-m}}\right)$$

$$+ \rho^{t_k - t_{k-m-1} - n_{t_k, t_{k-m-1}}^k} \left(\frac{M(\mu + \lambda)}{\mu(M + \lambda)}\right)^m \left(1 - \rho^{n_{t_{k-m}, t_{k-m-1}+1}^k - n_{t_{k-m}, t_{k-m-1}+1}^{k-m}}\right)$$

$$\frac{M(\mu + \lambda)}{\mu(M + \lambda)} \rho^{t_{k-m-1} - s - n_{t_{k-m-1}, s}^{k-m}} \left(1 - \rho^{n_{t_{k-m-1}, s+1}^{k-m} - n_{t_{k-m-1}, s+1}^{k-m}}\right)$$

$$\leq \rho^{t_k - s - n_{t_k, s}^k} \left(\frac{M(\mu + \lambda)}{\mu(M + \lambda)}\right)^{m+1} \left(1 - \rho^{n_{t_{k-m-1}, s+1}^k - n_{t_{k-m-1}, s+1}^{k-m}}\right)$$

$$+ \rho^{t_k - s - n_{t_k, t_{k-m-1}}^k - n_{t_{k-m-1}, s}^{k-m}} \left(\frac{M(\mu + \lambda)}{\mu(M + \lambda)}\right)^{m+1} \left(1 - \rho^{n_{t_{k-m-1}, s+1}^{k-m} - n_{t_{k-m-1}, s+1}^{k-m}}\right)$$

$$\leq \rho^{t_k - s - n_{t_k, s}^k} \left(\frac{M(\mu + \lambda)}{\mu(M + \lambda)}\right)^{m+1} \left(1 - \rho^{n_{t_{k-m-1}, s+1}^k - n_{t_{k-m-1}, s+1}^{k-m}}\right)$$

$$\leq \rho^{t_k - s - n_{t_k, s}^k} \left(\frac{M(\mu + \lambda)}{\mu(M + \lambda)}\right)^{m+1} \left(1 - \rho^{n_{t_{k-m-1}, s+1}^k - n_{t_{k-m-1}, s+1}^{k-m-1}}\right),$$

where in the last line we utilize $n_{t_{k-m-1}, s+1}^{k-m} > n_{t_{k-m-1}, s+1}^{k-m-1}$ to complete the proof. □

*proof of Lemma D.1.* We aim to upper bound the approximation error $\|w_{t_k}^{-\mathbf{S}_{1:t_k}} - w_{t_k}^{-S_{\leq t_k}}\|$, from which the result can be naturally extended to $\|w_t^{-\mathbf{S}_{1:t}} - w_t^{-S_{\leq t}}\|$. To this end, we first rewrite $w_{t_k}^{-\mathbf{S}_{1:t_k}} - w_{t_k}^{-S_{\leq t_k}}$ based on Lemma D.4. Note that under the iterative update in (13), each $\bar{\Delta}_t$ is a linear combination of $\Delta_s$ for all $s \in S_{\leq t}$. Hence, define an operator $\pi_s(\cdot)$ that mapps $w_{t_k}^{-\mathbf{S}_{1:t_k}} - w_{t_k}^{-S_{\leq t_k}}$ to the term only have $\Delta_s$, we can rewrite $w_{t_k}^{-\mathbf{S}_{1:t_k}} - w_{t_k}^{-S_{\leq t_k}}$ as:

$$w_{t_k}^{-\mathbf{S}_{1:t_k}} - w_{t_k}^{-S_{\leq t_k}} = \sum_{i=0}^{k-1} \sum_{s \in S_{t_{k-i}}} \pi_s(w_{t_k}^{-\mathbf{S}_{1:t_k}} - w_{t_k}^{-S_{\leq t_k}})\Delta_s = \sum_{i=0}^{k-1} \sum_{s \in S_{t_{k-i}}} D_j(t_k, s)\Delta_s,$$

where we define $\pi_s(w_{t_k}^{-\mathbf{S}_{1:t_k}} - w_{t_k}^{-S_{\leq t_k}}) = D_j(t_k, s)$, for $s \in S_{t_{k-j}}$. It is easy to show the following equations by Lemma D.4:

$$D_0(t_k, s) = \prod_{\substack{i=s+1 \\ i \notin S_{\leq t_k}}}^{t_k} \tilde{P}_i - \prod_{\substack{i=s+1 \\ i \notin S_{\leq t_k}}}^{t_k} P_i,$$

$$D_{j+1}(t_k, s) = D_j(t_k, s) - D_j(t_k, t_{k-j-1}) \prod_{\substack{i=s+1 \\ i \notin S_{\leq t_{k-j-1}}}}^{t_{k-j-1}} P_i. \tag{20}$$

Therefore, we can upper bound the approximation error as

$$\|w_{t_k}^{-\mathbf{S}_{1:t_k}} - w_{t_k}^{-S_{\leq t_k}}\| = \|\sum_{i=0}^{k-1} \sum_{s \in S_{t_{k-i}}} D_j(t_k, s)\Delta_s\| \leq \sum_{i=0}^{k-1} \sum_{s \in S_{t_{k-i}}} \|D_j(t_k, s)\|\frac{L}{\lambda},$$

where the $\|\Delta_s\| \leq \frac{L}{\lambda}$ is given by the first order condition under algorithm 1 and the $L$-Lipschitz condition.

To upper bound the remaining part $\|D_j(t_k, s)\|$, first note that

$$D_{j+1}(t_k, s)$$

$$=D_j(t_k, s) - D_j(t_k, t_{k-j-1}) \prod_{\substack{i=s+1 \\ i \notin S_{\leq t_{k-j-1}}}}^{t_{k-j-1}} P_i$$

$$=D_j(t_k, s) - D_j(t_k, t_{k-j-1}) \prod_{\substack{i=s+1 \\ i \notin S_{\leq t_{k-j}}}}^{t_{k-j-1}} P_i + D_j(t_k, t_{k-j-1}) \prod_{\substack{i=s+1 \\ i \notin S_{\leq t_{k-j}}}}^{t_{k-j-1}} P_i$$

$$- D_j(t_k, t_{k-j-1}) \prod_{\substack{i=s+1 \\ i \notin S_{\leq t_{k-j-1}}}}^{t_{k-j-1}} P_i$$

$$=\left(D_{j-1}(t_k, s) - D_{j-1}(t_k, t_{k-j-1}) \prod_{\substack{i=s+1 \\ i \notin S_{\leq t_{k-j}}}}^{t_{k-j-1}} P_i\right) + D_j(t_k, t_{k-j-1}) \left( \prod_{\substack{i=s+1 \\ i \notin S_{\leq t_{k-j}}}}^{t_{k-j-1}} P_i - \prod_{\substack{i=s+1 \\ i \notin S_{\leq t_{k-j-1}}}}^{t_{k-j-1}} P_i \right),$$

where the last equation is derived by further expanding $D_j(t_k, s)$, $D_j(t_k, t_{k-j-1})$ and reranging. In the last line, the term in the first bracket has the same form to $D_j(t_k, t_{k-j-1}) = D_{j-1}(t_k, s) - D_{j-1}(t_k, t_{k-j}) \prod_{i=s+1, i \in S_{\leq t_{k-j}}}^{t_{k-j}} P_i$ with only different on the $t_{k-j-1}$, thus we denote it as $D'_j(t_k, s)$. Then by Lemma D.3, we have

$$\|D_{j+1}(t_k, s)\|$$

$$\leq \|D'_j(t_k, s)\| + \|D_j(t_k, s)\| \frac{M(\mu+\lambda)}{(M+\lambda)\mu} \rho^{t_{k-j-1}-s-n_{t_{k-j-1},s+1}^{k-j-1}} (1 - \rho^{n_{t_{k-j},s+1}^{k-j-1} - n_{t_{k-j-1},s+1}^{k-j-1}}). \tag{21}$$

When $j = 0$, the upper bound for $D_0(t_k, s)$ and $D'_0(t_k, s)$ are both

$$(t_k - s - n_{t_k,s+1}^k) \rho^{t_k - s - n_{t_k,s+1}^k} \frac{M-\mu}{\mu+\lambda}. \tag{22}$$

by Lemma D.2. Thus, iteratively expanding (21) and substituting the definition of parameter $C$, together with the upper bound $L/\lambda$ for all $\|\Delta_s\|$, we can obtain the results.

**Lemma D.2.** *Fix time $t$. Define*

$$\tilde{H}_i = \int_0^1 \nabla^2 \tilde{F}_i \left( w_i^{-\mathbf{S}_{1:i-1}} + u \left( w_i^{-S_{\leq t} \setminus \{i+1, \dots, t\}} - w_i^{-\mathbf{S}_{1:i}} \right) \right) du,$$

$$H_i = \nabla^2 \tilde{F}_i \left( w_i^{-\mathbf{S}_{1:i-1}} \right), \quad \tilde{P}_i = (\tilde{H}_i + \lambda I)^{-1} \lambda I, \quad P_i = (H_i + \lambda I)^{-1} \lambda I$$

*. Then we have*

$$\prod_{i=s}^t \tilde{P}_i - \prod_{i=s}^t P_i \leq (t - s + 1) \rho^{t-s+1} \frac{M-\mu}{\mu+\lambda}.$$

*proof of Lemma D.2.* By the telescoping, we have

$$\|\prod_{i=s}^t \tilde{P}_i - \prod_{i=s}^t P_i\| = \|(\tilde{P}_t - P_t) \prod_{i=s}^{t-1} \tilde{P}_i + P_t(\tilde{P}_{t-1} - P_{t-1}) \prod_{i=s}^{t-2} \tilde{P}_i + \cdots + \prod_{i=s+1}^t P_i(\tilde{P}_s - P_s)\|. \tag{23}$$

$\|\tilde{P}_i\|$ and $\|P_i\|$ are all upper bounded by $\rho$. For any $\|\tilde{P}_i - P_i\|$, we have

$$\|\tilde{P}_i - P_i\| = \lambda \|(\tilde{H}_i + \lambda I)^{-1}(H_i - \tilde{H}_i)(H_i + \lambda I)^{-1}\| \leq \frac{\lambda(M-\mu)}{(\mu+\lambda)^2}.$$

Applying the upper bound for $\|\tilde{P}_i - P_i\|$, $\|\tilde{P}_i\|$ and $\|P_i\|$ in (23), we complete the proof. $\square$

**Lemma D.3.** *Define $H_i = \nabla^2 \tilde{F}_i\left(w_i^{-\mathbf{S}_{1:i-1}}\right)$ and $P_i = (H_i + \lambda I)^{-1}\lambda I$. Then we have the following inequalities*

$$
\left( \prod_{\substack{i=s+1 \\ i \notin S_{\leq t_{k-j}}}}^{t_{k-j-1}} P_i - \prod_{\substack{i=s+1 \\ i \notin S_{\leq t_{k-j-1}}}}^{t_{k-j-1}} P_i \right) \leq \frac{M(\mu+\lambda)}{(M+\lambda)\mu} \rho^{t_{k-j-1}-s-n^{k-j-1}_{t_{k-j-1},s+1}} \left(1 - \rho^{n^{k-j-1}_{t_{k-j},s+1}-n^{k-j-1}_{t_{k-j-1},s+1}}\right)
$$

*Proof.* Since $S_{\leq t_{k-j}} = S_{\leq t_{k-j-1}} \cup S_{t_{k-j}}$, compared with the second product, the first one excludes an additional set of indices

$$
R := S_{t_{k-j}} \cap [s+1, t_{k-j-1}].
$$

Interpreting "excluding $i$" as replacing $P_i$ by $I$, define

$$
Q^{(0)} = \prod_{\substack{i=s+1 \\ i \notin S_{\leq t_{k-j-1}}}}^{t_{k-j-1}} P_i, \qquad Q^{(*)} = \prod_{\substack{i=s+1 \\ i \notin S_{\leq t_{k-j}}}}^{t_{k-j-1}} P_i.
$$

If $R = \emptyset$, the difference vanishes. Otherwise, let $R = \{r_1 < \cdots < r_m\}$. For $\ell = 0, \ldots, m$, define

$$
Q^{(\ell)} := Q^{(0)} \text{ with } P_{r_1}, \ldots, P_{r_\ell} \text{ replaced by } I.
$$

Clearly $Q^{(m)} = Q^{(*)}$, and

$$
Q^{(*)} - Q^{(0)} = \sum_{\ell=1}^{m} \left( Q^{(\ell)} - Q^{(\ell-1)} \right).
$$

For any $\ell$. Only the factor at position $r_\ell$ changes, so

$$
\|Q^{(\ell)} - Q^{(\ell-1)}\| \leq \|I - P_{r_\ell}\| \rho^{t_{k-j-1}-s-n^{k-j-1}_{t_{k-j-1},s+1}+l-1},
$$

where $\rho$ serves as the upper bound of any $P_i$. Since $I - P_i = (H_i + \lambda I)^{-1}H_i$, we have $\|I - P_i\| < \frac{M}{M+\lambda}$ according to the assumption of $M$-smoothness. Therefore, we have

$$
\begin{aligned}
\sum_{\ell=1}^{m} \left( Q^{(\ell)} - Q^{(\ell-1)} \right) &\leq \frac{M}{M+\lambda} \sum_{\ell=1}^{m} \rho^{t_{k-j-1}-s-n^{k-j-1}_{t_{k-j-1},s+1}+l-1} \\
&\leq \frac{M}{M+\lambda} \rho^{t_{k-j-1}-s-n^{k-j-1}_{t_{k-j-1},s+1}} \frac{1-\rho^m}{1-\rho} \\
&= \frac{M(\mu+\lambda)}{(M+\lambda)\mu} \rho^{t_{k-j-1}-s-n^{k-j-1}_{t_{k-j-1},s+1}} \left(1 - \rho^{n^{k-j-1}_{t_{k-j},s+1}-n^{k-j-1}_{t_{k-j-1},s+1}}\right),
\end{aligned}
$$

then we complete the proof.

$\square$

$\square$

**Lemma D.4.** *Fix time $t$. Define*

$$
\tilde{H}_i = \int_0^1 \nabla^2 \tilde{F}_i\left(w_i^{-\mathbf{S}_{1:i-1}} + u\left(w_i^{-S_{\leq t}\setminus\{i+1,\ldots,t\}} - w_i^{-\mathbf{S}_{1:i}}\right)\right) du,
$$

$$
H_i = \nabla^2 \tilde{F}_i\left(w_i^{-\mathbf{S}_{1:i-1}}\right), \quad \tilde{P}_i = (\tilde{H}_i + \lambda I)^{-1}\lambda I, \quad P_i = (H_i + \lambda I)^{-1}\lambda I.
$$

*Then, for the unlearning model $w_t^{-\mathbf{S}_{1:t}}$ updated in Alg. 1 and the retraining model $w_t^{-S_{\leq t}}$, the following equation holds:*

$$
\begin{aligned}
&w_t^{-S_{\leq t}} - w_t^{-\mathbf{S}_{1:t}} \\
&= \sum_{s \in S_{\leq t}} \left( \prod_{i=s+1, i \notin S_{\leq t}}^{t} \tilde{P}_i - \prod_{i=s+1, i \notin S_{\leq t}}^{t} P_i \right) \Delta_s - \sum_{\tau \in U_{t-1}} \left( \prod_{i=\tau+1, i \notin S_{\leq t}}^{t} \tilde{P}_i - \prod_{i=\tau+1, i \notin S_{\leq t}}^{t} P_i \right) \bar{\Delta}_\tau
\end{aligned}
$$

(24)

*Proof of Lemma D.4.* Under the continual learning algorithm (1), we have the following first-order conditions for the unlearning model $w_t^{-\mathbf{S}_{1:t}}$ and retraining model $w_t^{-S_{\leq t}}$:

$$\nabla \tilde{F}_t(w_t^{-S_{\leq t}}) + \lambda(w_t^{-S_{\leq t}} - w_{t-1}^{-S_{\leq t}}) = 0, \ \nabla \tilde{F}_t(w_t^{-\mathbf{S}_{1:t-1}}) + \lambda(w_t^{-\mathbf{S}_{1:t-1}} - w_{t-1}^{-\mathbf{S}_{1:t-1}}) = 0.$$

Having $\nabla \tilde{F}_t(w_t^{-S_{\leq t}})$ Taylor expansion at the point of $w_t^{-\mathbf{S}_{1:t}}$, we have

$$\tilde{F}_t(w_t^{-\mathbf{S}_{1:t-1}}) + \tilde{H}_t(w_t^{-S_{\leq t}} - w_t^{-\mathbf{S}_{1:t-1}}) + \lambda(w_t^{-S_{\leq t}} - w_{t-1}^{-S_{\leq t}}) = 0$$

$$\implies -\lambda(w_t^{-\mathbf{S}_{1:t-1}} - w_{t-1}^{-\mathbf{S}_{1:t-1}}) + \tilde{H}_t(w_t^{-S_{\leq t}} - w_t^{-\mathbf{S}_{1:t-1}}) + \lambda(w_t^{-S_{\leq t}} - w_{t-1}^{-S_{\leq t}}) = 0$$

$$\implies w_t^{-S_{\leq t}} - w_t^{-\mathbf{S}_{1:t-1}} = (\tilde{H}_t + \lambda I)^{-1} \lambda (w_{t-1}^{-S_{\leq t}} - w_{t-1}^{-\mathbf{S}_{1:t-1}}), \tag{25}$$

We will first prove the following claim for any $t = 1, \ldots, T$:

$$w_t^{-S_{\leq t}} - w_t^{-\mathbf{S}_{1:t-1}} = \sum_{s \in S_{\leq t}} \prod_{i=s+1, i \notin S_{\leq t}}^{t} \tilde{P}_i \Delta_s - \sum_{\tau \in U_{t-1}} \prod_{i=\tau+1, i \notin S_{\leq t}}^{t} \tilde{P}_i \bar{\Delta}_\tau. \tag{26}$$

To show this, we will use induction to prove the following claim:

$$w_\tau^{-S_{\leq t} \setminus \{\tau+1, \ldots, t\}} - w_\tau^{-\mathbf{S}_{1:\tau-1}} = \sum_{s \in S_{\leq t} \setminus \{\tau+1, \ldots, t\}} \prod_{i=s+1, i \notin S_{\leq t}}^{\tau} \tilde{P}_i \Delta_s - \sum_{j \in U_{\tau-1}} \prod_{i=j+1, i \notin S_{\leq t}}^{\tau} \tilde{P}_i \bar{\Delta}_\tau$$
$$\tag{27}$$

Once we prove the equation in (27), (26) holds under $\tau = t$.

First, for $\tau < \min S_{\leq t}$, we have

$$w_\tau^{-S_{\leq \tau} \setminus \{\tau+1, \ldots, t\}} - w_\tau^{-\mathbf{S}_{1:\tau-1}} = \mathbf{0},$$

since both $S_{\leq t} \setminus \{\tau + 1, \ldots, t\}$ and $U_{\tau-1}$ are empty. This is because, at time $\tau$, it is impossible to request unlearning of any future task in $S_{\leq t}$.

At time $\tau$, suppose the equation (27) holds. Then, at time $\tau + 1$, if $\tau + 1 \in S_{\leq t}$, we have

$$w_{\tau+1}^{-S_{\leq t} \setminus \{\tau+2, \ldots, t\}} - w_{\tau+1}^{-\mathbf{S}_{1:\tau}}$$

$$= w_\tau^{-S_{\leq t} \setminus \{\tau+1, \ldots, t\}} - w_{\tau+1}^{-\mathbf{S}_{1:\tau}}$$

$$= w_\tau^{-S_{\leq t} \setminus \{\tau+1, \ldots, t\}} - w_\tau^{-\mathbf{S}_{1:\tau}} + w_\tau^{-\mathbf{S}_{1:\tau}} - w_{\tau+1}^{-\mathbf{S}_{1:\tau}}$$

$$= w_\tau^{-S_{\leq t} \setminus \{\tau+1, \ldots, t\}} - w_\tau^{-\mathbf{S}_{1:\tau-1}} + \Delta_{\tau+1} - \bar{\Delta}_\tau$$

$$= \sum_{s \in S_{\leq t} \setminus \{\tau+1, \ldots, t\}} \prod_{i=s+1, i \notin S_{\leq t}}^{\tau} \tilde{P}_i \Delta_s - \sum_{j \in U_{\tau-1}} \prod_{i=j+1, i \notin S_{\leq t}}^{\tau} \tilde{P}_i \bar{\Delta}_\tau + \Delta_{\tau+1} - \bar{\Delta}_\tau$$

$$= \sum_{s \in S_{\leq t} \setminus \{\tau+2, \ldots, t\}} \prod_{i=s+1, i \notin S_{\leq t}}^{\tau+1} \tilde{P}_i \Delta_s - \sum_{j \in U_\tau} \prod_{i=j+1, i \notin S_{\leq t}}^{\tau+1} \tilde{P}_i \bar{\Delta}_\tau,$$

where the last two equations hold because $\tau + 1 \in S_{\leq t}$, and if $\tau$ is not an unlearning time, then $\bar{\Delta}_\tau = \mathbf{0}$; otherwise, $\tau \cup U_{\tau-1} \in U_\tau$.

If $\tau + 1 \notin S_{\leq t}$, by (25), we have the following:

$$w_{\tau+1}^{-S_{\leq t} \setminus \{\tau+2, \ldots, t\}} - w_{\tau+1}^{-\mathbf{S}_{1:\tau}} =$$

$$\tilde{P}_{\tau+1}(w_\tau^{-S_{\leq t} \setminus \{\tau+2, \ldots, t\}} - w_\tau^{-\mathbf{S}_{1:\tau}}) = \tilde{P}_{\tau+1}(w_\tau^{-S_{\leq t} \setminus \{\tau+1, \ldots, t\}} - w_\tau^{-\mathbf{S}_{1:\tau-1}} - \bar{\Delta}_\tau),$$

where $\bar{\Delta}_\tau = \mathbf{0}$ if $\tau \notin U_t$ is not an unlearning time. Following this, we consider the following two cases to complete the induction for (27)

- If $\tau \in S_{\leq t}$, we have $w_\tau^{-S_{\leq t}\backslash\{\tau+1,...,t\}} = w_{\tau-1}^{-S_{\leq t}\backslash\{\tau,\tau+1,...,t\}}$, then

$$
w_{\tau+1}^{-S_{\leq t}\backslash\{\tau+2,...,t\}} - w_{\tau+1}^{-\mathbf{S}_{1:\tau}}
$$

$$
= \tilde{P}_{\tau+1}(w_\tau^{-S_{\leq t}\backslash\{\tau+1,...,t\}} - w_\tau^{-\mathbf{S}_{1:\tau-1}} - \bar{\Delta}_\tau)
$$

$$
= \tilde{P}_{\tau+1}(w_{\tau-1}^{-S_{\leq t}\backslash\{\tau,\tau+1,...,t\}} - w_\tau^{-\mathbf{S}_{1:\tau-1}} - \bar{\Delta}_\tau)
$$

$$
= \tilde{P}_{\tau+1}(w_{\tau-1}^{-S_{\leq t}\backslash\{\tau,\tau+1,...,t\}} - w_{\tau-1}^{-\mathbf{S}_{1:\tau-1}} + w_{\tau-1}^{-\mathbf{S}_{1:\tau-1}} - w_\tau^{-\mathbf{S}_{1:\tau-1}} - \bar{\Delta}_\tau)
$$

$$
= \tilde{P}_{\tau+1}(w_{\tau-1}^{-S_{\leq t}\backslash\{\tau,\tau+1,...,t\}} - w_{\tau-1}^{-\mathbf{S}_{1:\tau-2}} - \bar{\Delta}_{\tau-1} - \bar{\Delta}_\tau) + \tilde{P}_{\tau+1}(w_{\tau-1}^{-\mathbf{S}_{1:\tau-1}} - w_\tau^{-\mathbf{S}_{1:\tau-1}})
$$

$$
= \sum_{s \in S_{\leq t}\backslash\{\tau,...,t\}} \prod_{i=s+1, i\notin S_{\leq t}}^{\tau+1} \tilde{P}_i \Delta_s - \sum_{j \in U_{\tau-2}} \prod_{i=j+1, i\notin S_{\leq t}}^{\tau+1} \tilde{P}_i \bar{\Delta}_\tau + \tilde{P}_{\tau+1}\Delta_\tau
$$

$$
- \tilde{P}_{\tau+1}\bar{\Delta}_{\tau-1} - \tilde{P}_{\tau+1}\bar{\Delta}_\tau
$$

$$
= \sum_{s \in S_{\leq t}\backslash\{\tau+2,...,t\}} \prod_{i=s+1, i\notin S_{\leq t}}^{\tau+1} \tilde{P}_i \Delta_s - \sum_{j \in U_\tau} \prod_{i=j+1, i\notin S_{\leq t}}^{\tau+1} \tilde{P}_i \bar{\Delta}_\tau.
$$

- If $\tau \notin S_{\leq t}$, we have $w_\tau^{-S_{\leq t}\backslash\{\tau+1,...,t\}} = w_\tau^{-S_{\leq t}\backslash\{\tau,\tau+1,...,t\}}$, then

$$
w_{\tau+1}^{-S_{\leq t}\backslash\{\tau+2,...,t\}} - w_{\tau+1}^{-\mathbf{S}_{1:\tau}}
$$

$$
= \tilde{P}_{\tau+1}(w_\tau^{-S_{\leq t}\backslash\{\tau+1,...,t\}} - w_\tau^{-\mathbf{S}_{1:\tau-1}} - \bar{\Delta}_\tau)
$$

$$
= \sum_{s \in S_{\leq t}\backslash\{\tau,...,t\}} \prod_{i=s+1, i\notin S_{\leq t}}^{\tau+1} \tilde{P}_i \Delta_s - \sum_{j \in U_{\tau-1}} \prod_{i=j+1, i\notin S_{\leq t}}^{\tau+1} \tilde{P}_i \bar{\Delta}_\tau - \tilde{P}_{\tau+1}\bar{\Delta}_\tau
$$

$$
= \sum_{s \in S_{\leq t}\backslash\{\tau+2,...,t\}} \prod_{i=s+1, i\notin S_{\leq t}}^{\tau+1} \tilde{P}_i \Delta_s - \sum_{j \in U_\tau} \prod_{i=j+1, i\notin S_{\leq t}}^{\tau+1} \tilde{P}_i \bar{\Delta}_\tau.
$$

Then we complete the proof for (26).

Then, combining (26) and the update of $w_t^{-\mathbf{S}_{1:t}}$ in (13), we complete the proof.

## D.2 Proof of Proposition 5.2

By (25), we have

$$
\|w_t^{-S_{\leq t}} - w_t^{-\mathbf{S}_{1:t-1}}\| = \|\prod_{i=t_k+1}^{t} P_i\big(w_{t_k}^{-S_{\leq t_k}} - w_{t_k}^{-\mathbf{S}_{1:t_k}}\big)\| \leq \rho^{t-t_k}\|w_{t_k}^{-S_{\leq t_k}} - w_{t_k}^{-\mathbf{S}_{1:t_k}}\|,
$$

then we focus on upper bounding $G_k = \|w_{t_k}^{-S_{\leq t_k}} - w_{t_k}^{-\mathbf{S}_{1:t_k}}\|$. According to continual learning algorithm (1), we have the following first-order conditions:

$$
\nabla \tilde{F}_t(w_{t_k}^{-S_{\leq t_k}}) + \lambda(w_{t_k}^{-S_{\leq t_k}} - w_{t_k-1}^{-S_{\leq t_k}}) = 0, \ \nabla \tilde{F}_t(w_{t_k}^{-\mathbf{S}_{1:t_k-1}}) + \lambda(w_{t_k}^{-\mathbf{S}_{1:t_k-1}} - w_{t_k-1}^{-\mathbf{S}_{1:t_k-1}}) = 0.
$$

Having $\nabla \tilde{F}_t(w_t^{-S_{\leq t}})$ Taylor expansion to the third order at the point of $w_t$, we have

$$
\nabla^2 \tilde{F}_{t_k}(w_{t_k}^{-\mathbf{S}_{1:t_k-1}})
$$

$$
+ \int_0^1 (1-\tau)\left(\nabla^3 \tilde{F}_t(w_{t_k}^{-\mathbf{S}_{1:t_k-1}} + h(w_{t_k}^{-S_{\leq t_k}} - w_{t_k}^{-\mathbf{S}_{1:t_k-1}}))\right.
$$

$$
\left. - \nabla^3 \tilde{F}_t(w_{t_k}^{-\mathbf{S}_{1:t_k-1}} + h(w_{t_k}^{-S_{\leq t_k}} - w_{t_k}^{-\mathbf{S}_{1:t_k-1}}))\right)
$$

$$
[w_{t_k}^{-\mathbf{S}_{1:t_k-1}} + h(w_{t_k}^{-S_{\leq t_k}} - w_{t_k}^{-\mathbf{S}_{1:t_k-1}}), w_{t_k}^{-\mathbf{S}_{1:t_k-1}} + h(w_{t_k}^{-S_{\leq t_k}} - w_{t_k}^{-\mathbf{S}_{1:t_k-1}})] \, dh
$$

$$
+ \lambda(w_{t_k}^{-S_{\leq t_k}} - w_{t_k}^{-\mathbf{S}_{1:t_k-1}}) = \lambda(w_{t_k-1}^{-S_{\leq t_k}} - w_{t_k-1}^{-\mathbf{S}_{1:t_k-1}})
$$

Applying the $L_3$ Hessian Lipschitz condition, we have

$$\|w_{t_k}^{-S_{\le t_k}} - w_{t_k}^{-\mathbf{S}_{1:t_k}}\| = \|w_{t_k}^{-S_{\le t_k}} - w_{t_k}^{-\mathbf{S}_{1:t_k-1}} - \bar{\Delta}_{t_k}\|$$

$$\le \|P_{t_k}(w_{t_k-1}^{-S_{\le t_k}} - w_{t_k-1}^{-\mathbf{S}_{1:t_k-1}}) - \bar{\Delta}_{t_k}\| + \frac{L_3}{2}\|w_{t_k}^{-S_{\le t_k}} - w_{t_k}^{-\mathbf{S}_{1:t_k-1}}\|^2$$

$$\le \|\prod_{i=s_n+1}^{t_k} P_i(w_{s_n}^{-S_{\le t_k}} - w_{s_n}^{-\mathbf{S}_{1:t_k-1}}) - \bar{\Delta}_{t_k}\|$$

$$+ \sum_{m=s_n+1}^{t_k} \frac{L_3}{2}\|(w_m^{-S_{\le t_k}} - w_m^{-\mathbf{S}_{1:t_k-1}})\|^2,$$

where $s_n$ represents the largest task in $S_{\le t}$. For the first term in the last line, we have

$$\|\prod_{i=s_n+1}^{t_k} P_i(w_{s_n}^{-S_{\le t_k}} - w_{s_n}^{-\mathbf{S}_{1:t_k-1}}) - \bar{\Delta}_{t_k}\|$$

$$= \|\prod_{i=s_n+1}^{t_k} P_i(w_{s_n-1}^{-S_{\le t_k}\backslash\{s_n\}} - w_{s_n}^{-\mathbf{S}_{1:t_k-1}}) - \bar{\Delta}_{t_k}\|$$

$$= \|\prod_{i=s_n+1}^{t_k} P_i(w_{s_n-1}^{-S_{\le t_k}\backslash\{s_n\}} - w_{s_n-1}^{-\mathbf{S}_{1:t_k-1}}) + \prod_{i=s_n+1}^{t_k} P_i\Delta_{s_n} - \bar{\Delta}_{t_k}\|$$

In the last line, $\prod_{i=s_n+1}^{t_k} P_i\Delta_{s_n}$ is a term of $\bar{\Delta}_{t_k}$, and thus it cancels out. Similarly, if we iterate to $w_{t_i}^{-\mathbf{S}_{1:t_i}} = w_{t_i}^{-\mathbf{S}_{1:t_i}} - \bar{\Delta}_{t_i}$, it will also canels out with the term in $\bar{\Delta}_{t_k}$. Keep the iteration above then we complete the proof.

$\square$

### D.3 Performance guarantees of the forgetting enhanced Hessian-based unlearning algorithm

We prove the performance guarantee of this modified Alg. 2 below.

**Theorem D.5.** *For each $i$-th unlearning request $t_i \in U_t = \{t_1, \ldots, t_k\}$, let $n_{t_i, s+1}^i$ denote the number of tasks in time interval $[s + 1, t_i]$ that have been deleted by time $t_i$. The modified Alg. 2 using noise coefficient $\gamma_t(\mathbf{S}_{1:t})$ below tells unlearning loss and guarantees $(\varepsilon, \delta)$-certified continual unlearning in Definition 2.1:*

$$\gamma_t(\mathbf{S}_{1:t}) :=$$

$$\sum_{i=1}^{k} \sum_{s \in S'_{t_i}} (t_i - s - n_{t_i, s+1}^i)\rho^{t-s-n_{t_i, s+1}^i}\frac{L(M-\mu)}{\lambda(\mu+\lambda)} + \sum_{i=2}^{k} \sum_{s \in S_{t_i}\backslash S'_{t_i}} \rho^{t-s-n_{t_i, s+1}^i}\frac{L}{\lambda}. \quad (28)$$

*Further, the output model $\tilde{w}_t^{-\mathbf{S}_{1:t}}$ of modified Alg. 2 achieves the post-unlearning excess risk upper bound as defined in Definition 2.2 of $L\left(\sqrt{2d\ln(\frac{1.25}{\delta})}/\varepsilon+1\right)\gamma_t(\mathbf{S}_{1:t})+\mathcal{E}^{-S_{\le t}}(\lambda)$, where $\mathcal{E}^{-S_{\le t}}(\lambda)$ is given in (8), and $\gamma_t(\mathbf{S}_{1:t})$ is in (28).*

*proof of Lemma 5.4.* We will prove by induction. At time $t$, the claim hold since $U_{t_1-1} = \emptyset$. At unlearning time $t_k$, suppose the claim holds as

$$\bar{\Delta}_{t_k} = \sum_{s \in S_{t_k}} \prod_{i=s+1, i \notin S_{\le t_k}}^{t_k} (H_i + \lambda I)^{-1}\lambda\Delta_s$$

Then, at the next unlearning time $t_{k+1}$, we have the following:

$$\bar{\Delta}_{t_{k+1}} = \sum_{s \in S_{t_{k+1}}} \prod_{i=s+1, i \notin S_{t_{k+1}}}^{t_{k+1}} (H_i + \lambda I)^{-1} \lambda \Delta_s +$$

$$\Big( \sum_{s \in S_{\leq t_k}} \prod_{i=s+1, i \notin S_{\leq t_{k+1}}}^{t_{k+1}} (H_i + \lambda I)^{-1} \lambda \Delta_s - \sum_{i=1}^{k} \prod_{i=t_i+1, i \notin S_{\leq t_{k+1}}}^{t_{k+1}} (H_i + \lambda I)^{-1} \lambda \bar{\Delta}_{t_i} \Big) \quad (29)$$

$$= \sum_{s \in S_{t_{k+1}}} \prod_{i=s+1, i \notin S_{t_{k+1}}}^{t_{k+1}} (H_i + \lambda I)^{-1} \lambda \Delta_s +$$

$$\Big( \sum_{s \in S_{\leq t_k}} \prod_{i=s+1, i \notin S_{\leq t_{k+1}}}^{t_{k+1}} (H_i + \lambda I)^{-1} \lambda \Delta_s - \sum_{i=1}^{k-1} \prod_{i=t_i+1, i \notin S_{\leq t_{k+1}}}^{t_{k+1}} (H_i + \lambda I)^{-1} \lambda \bar{\Delta}_{t_i} \Big) \quad (30)$$

$$- \prod_{i=t_k+1, i \notin S_{\leq t_{k+1}}}^{t_{k+1}} (H_i + \lambda I)^{-1} \lambda \bar{\Delta}_{t_k} \quad (31)$$

Since for all $s \in S_{t_{k+1}}, s \geq t_k$, we can write the following equations for any $s \in S_{\leq t_k}$

$$\prod_{i=s+1, i \notin S_{\leq t_{k+1}}}^{t_{k+1}} (H_i + \lambda I)^{-1} \lambda = \prod_{i=t_k+1, i \notin S_{t_{k+1}}}^{t_{k+1}} (H_i + \lambda I)^{-1} \lambda \prod_{i=s+1, i \notin S_{\leq t_k}}^{t_k} (H_i + \lambda I)^{-1} \lambda$$

$$\prod_{i=t_i+1, i \notin S_{\leq t_{k+1}}}^{t_{k+1}} (H_i + \lambda I)^{-1} \lambda = \prod_{i=t_k+1, i \notin S_{\ t_{k+1}}}^{t_{k+1}} (H_i + \lambda I)^{-1} \lambda \prod_{i=t_i+1, i \notin S_{\leq t_k}}^{t_k} (H_i + \lambda I)^{-1} \lambda.$$

Then, we can rewrite (31) as

$$\sum_{s \in S_{t_{k+1}}} \prod_{i=s+1, i \notin S_{t_{k+1}}}^{t_{k+1}} (H_i + \lambda I)^{-1} \lambda \Delta_s + \prod_{i=t_k+1, i \notin S_{\ t_{k+1}}}^{t_{k+1}} (H_i + \lambda I)^{-1} \lambda \cdot$$

$$\Big( \sum_{s \in S_{\leq t_k}} \prod_{i=s+1, i \notin S_{\leq t_{k+1}}}^{t_k} (H_i + \lambda I)^{-1} \lambda \Delta_s - \sum_{i=1}^{k-1} \prod_{i=t_i+1, i \notin S_{\leq t_{k+1}}}^{t_k} (H_i + \lambda I)^{-1} \lambda \bar{\Delta}_{t_i} \Big)$$

$$- \prod_{i=t_k+1, i \notin S_{\leq t_{k+1}}}^{t_{k+1}} (H_i + \lambda I)^{-1} \lambda \bar{\Delta}_{t_k}$$

Since $\{i \in S_{t_{k+1}} : i < t_k\} = S_{t_k}$, the bracket in the second line equals $\bar{\Delta}_{t_k}$. This completes the induction. $\qquad\square$

*Proof of Theorem D.5.* Under the continual learning algorithm (1), we have the following first-order conditions for the unlearning model $w_t^{-\mathbf{S}_{1:t}}$ and retraining model $w_t^{-S_{\leq t}}$:

$$\nabla \tilde{F}_t(w_t^{-S_{\leq t}}) + \lambda(w_t^{-S_{\leq t}} - w_{t-1}^{-S_{\leq t}}) = 0, \ \nabla \tilde{F}_t(w_t^{-\mathbf{S}_{1:t-1}}) + \lambda(w_t^{-\mathbf{S}_{1:t-1}} - w_{t-1}^{-\mathbf{S}_{1:t-1}}) = 0.$$

Having $\nabla \tilde{F}_t(w_t^{-S_{\leq t}})$ Taylor expansion at the point of $w_t^{-\mathbf{S}_{1:t}}$, we have

$$\tilde{F}_t(w_t^{-\mathbf{S}_{1:t-1}}) + \tilde{H}_t(w_t^{-S_{\leq t}} - w_t^{-\mathbf{S}_{1:t-1}}) + \lambda(w_t^{-S_{\leq t}} - w_{t-1}^{-S_{\leq t}}) = 0$$

$$\implies -\lambda(w_t^{-\mathbf{S}_{1:t-1}} - w_{t-1}^{-\mathbf{S}_{1:t-1}}) + \tilde{H}_t(w_t^{-S_{\leq t}} - w_t^{-\mathbf{S}_{1:t-1}}) + \lambda(w_t^{-S_{\leq t}} - w_{t-1}^{-S_{\leq t}}) = 0$$

$$\implies w_t^{-S_{\leq t}} - w_t^{-\mathbf{S}_{1:t-1}} = (\tilde{H}_t + \lambda I)^{-1} \lambda (w_{t-1}^{-S_{\leq t}} - w_{t-1}^{-\mathbf{S}_{1:t-1}}) \quad (32)$$

$$= \prod_{i=t_k+1}^{t} (\tilde{H}_i + \lambda I)^{-1} \lambda (w_{t_k}^{-S_{\leq t_k}} - w_{t_k}^{-\mathbf{S}_{1:t_k}}), \quad (33)$$

Thus, $\|w_t^{-S_{\leq t}} - w_t^{-\mathbf{S}_{1:t-1}}\| \leq \rho^{t-t_k} \|w_{t_k}^{-S_{\leq t_k}} - w_{t_k}^{-\mathbf{S}_{1:t_k}}\|$, we focus on prove the upper bound of $\|w_{t_k}^{-S_{\leq t_k}} - w_{t_k}^{-\mathbf{S}_{1:t_k}}\|$. Applying the iteration between (33), and use the notation in Lemma D.2, we

have

$$w_{t_k}^{-S_{\leq t_k}} - w_{t_k}^{-\mathbf{S}_{1:t_k}} = w_{t_k}^{-S_{\leq t_k}} - w_{t_k}^{-\mathbf{S}_{1:t_{k-1}}} - \bar{\Delta}_{t_k}$$

$$= \sum_{s \in S'_{t_k}} \prod_{i=s+1, i \notin S'_{t_k}}^{t_k} \tilde{P} \Delta_s - \bar{\Delta}_{t_k}$$

$$+ \prod_{i=s+1}^{t_k} \prod_{i=s+1, i \notin S'_{t_k}}^{t_k} \tilde{P}(w_{t_k}^{-S_{\leq t_k} \setminus S'_{t_k}} - w_{t_{k-1}}^{-\mathbf{S}_{1:t_{k-1}}}).$$

Keeping this iteration, we will obtain

$$w_{t_k}^{-S_{\leq t_k}} - w_{t_k}^{-\mathbf{S}_{1:t_k}}$$

$$= \sum_{i=1}^{k} \sum_{s \in S'_{t_i}} \prod_{i=s+1, i \notin S_{\leq t_k}}^{t_k} (\tilde{P} - P) \Delta_s + \sum_{i=1}^{k} \sum_{s \in S_i \setminus S'_{t_i}} \prod_{i=s+1, i \notin S_{\leq t_i}}^{t_k} \tilde{P} \Delta_s.$$

By Lemma D.2, $\|\tilde{P}\| \leq \rho$ and $\|\Delta_s\| \leq L\lambda$, we can derive the results. $\qquad\square$

# E  ADDITIONAL EXPERIMENTS

## E.1  UNLEARNING LOSS VERSUS UNLEARNING SEQUENCE

We run the unlearning algorithms in Alg. 1 and Alg. 2 using a randomly generated unlearning sequence $t_1$ to $t_5$. To examine the impact of the unlearning sequence on the unlearning loss, we consider two sequence $S_{t_1} - S_{t_5}$ in the second and third rows of Table 2 to unlearn the same task dataset $\{1, 2, 7, 10, 12, 13, 16, 19, 23, 24\}$. The first sequence is well-ordered as in Lemma 5.4 , while in the second sequence, the unlearning requests for tasks 8, 13, 2, and 3 are advanced to earlier unlearning steps, thereby interrupting the original unlearning sequence. Fig. 3 (a)and 3 (b) show the approximation error results, which are proportional to the unlearning loss, under these two unlearning sequences.

In Fig. 3 (a) and 3 (b), each time the system receives an unlearning request, the approximation error jumps up and then decreases due to forgetting over time. Because the objective function is not strongly convex, continual learning exhibits limited natural forgetting. As a result, our Hessian-based algorithm consistently outperforms the natural-forgetting unlearning method. In particular, at time $t = 25$, when both models in the two sequences complete their ten unlearning tasks, the Hessian-based algorithm in Fig. 3 (a) yields a loss of 0.1, whereas in Fig. 3 (b) the loss increases to 0.13. This is because the unlearning sequence in Fig. 3 (b) arrives in an order where new requests disrupt the previous unlearning sequence.

| $t_i$ | 4 | 12 | 16 | 18 | 25 |
|-------|---|----|----|----|-----|
| $S_{t_i}$ | $1, 2$ | $7, 10$ | $12, 13$ | $16$ | $19, 23, 24$ |
| $S_{t_i}$ | $\emptyset$ | $10$ | $13$ | $7, 12, 16$ | $1, 2, 19, 23, 24$ |

Table 2: Two sequences of five unlearning requests' times and their deletion tasks

To examine the impact of the unlearning sequence on the unlearning loss, we consider two sequence $S_{t_1} - S_{t_5}$ in the second and third rows of Table 2 to unlearn the same task dataset $\{1, 2, 7, 10, 12, 13, 16, 19, 23, 24\}$. The first sequence is well-ordered as in Lemma 5.4 , while in the second sequence, the unlearning requests for tasks 8, 13, 2, and 3 are advanced to earlier unlearning steps, thereby interrupting the original unlearning sequence. Fig. 3 (a)and 3 (b) show the approximation error results, which are proportional to the unlearning loss, under these two unlearning sequences.

In Fig. 3 (a) and 3 (b), each time the system receives an unlearning request, the approximation error jumps up and then decreases due to forgetting over time. Because the objective function is not strongly convex, continual learning exhibits limited natural forgetting. As a result, our Hessian-based algorithm consistently outperforms the natural-forgetting unlearning method. In particular, at time $t = 25$, when both models in the two sequences complete their ten unlearning tasks, the Hessian-based algorithm in Fig. 3 (a) yields a loss of 0.1, whereas in Fig. 3 (b) the loss increases to 0.13. This is because the unlearning sequence in Fig. 3 (b) arrives in an order where new requests disrupt the previous unlearning sequence.

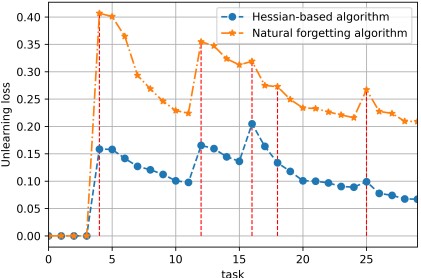 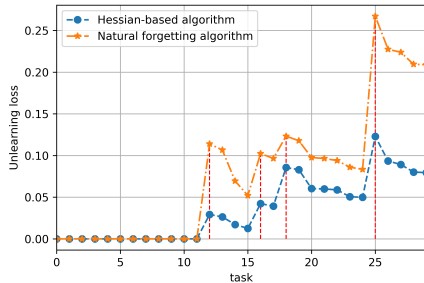

Figure 3: (a) Approximation errors for the first well-order unlearning sequence in Table 2. (b) Approximation errors for the second unlearning disruptive sequence Table 2.

