# OpenReview forum: "Machine Unlearning Meets Continuous Learning: A Theoretical Analysis"
_ICLR.cc/2026/Conference — Submitted to ICLR 2026_

### Official Review · Reviewer_hZQd · 2025-10-30

**Soundness:** 3
**Presentation:** 2
**Contribution:** 3
**Rating:** 6
**Confidence:** 3

**Summary:**

This paper builds the first theoretical bridge between machine unlearning and continual learning by formalizing an $(\varepsilon,\delta)$-certified continual unlearning setting in which tasks arrive sequentially and deletion requests may occur later, when original data are unavailable. The authors adopt $\ell_2$-regularized continual learning as the base learner, derive a new excess‑risk bound for it (extending prior linear results to nonlinear convex losses), and show how post‑unlearning performance decomposes into the continual‑learning excess risk and an unlearning loss.

**Strengths:**

1. The paper formulates the first certified continual unlearning setting that explicitly marries continual learning with $(\varepsilon,\delta)$-certified unlearning.
2. The paper extends excess‑risk bounds for $\ell_2$‑regularized continual learning from linear to nonlinear convex losses, derives explicit upper bounds with clear dependence on $\lambda$, task heterogeneity, and request sequences, and provides tight first‑ and second‑order approximation error bounds for unlearning.
3. This paper is fluently written with a detailed proof of the theorem.

**Weaknesses:**

1. Experiments. Only MNIST with a linear softmax model and 30 tasks.
2. Missing baselines. No experimental comparison to recent certified unlearning baselines adapted to continual settings.

**Questions:**

How do you practically calibrate the certification and noise—i.e., choose $(\varepsilon,\delta)$, estimate L/M/$\mu$ for deep models, and set $\lambda$—so that $\gamma_t(S_{1:t})$ is neither over- nor under-estimated, and how sensitive is post-unlearning excess risk to mis-specification of these constants? Could you provide a concrete recipe and ablation showing the impact of these choices?

---

### Official Review · Reviewer_xw6g · 2025-10-31

**Soundness:** 2
**Presentation:** 2
**Contribution:** 3
**Rating:** 4
**Confidence:** 3

**Summary:**

Briefly summarize the paper and its contributions. You can incorporate Markdown and Latex into your review. See https://openreview.net/faq.

This paper establishes a theoretical foundation connecting machine ****unlearning and continual learning. The authors formalize the problem of *continual learning–unlearning*, where a model must sequentially learn new tasks while also being able to efficiently and provably “forget” data upon deletion requests, without access to past datasets.

They adapt two classes of certified unlearning algorithms—(i) gradient-based and (ii) Hessian-based methods—into the continual learning setting. The framework provides rigorous performance guarantees by analyzing two coupled quantities: the excess risk arising from continual learning and the unlearning loss induced by the deletion process, whose balance determines the final post-unlearning excess risk.

Theoretical results show that the Hessian-based approach achieves smaller unlearning loss but at higher storage cost, while the gradient-based variant eliminates storage overhead by leveraging natural forgetting inherent in continual learning. Experimental results on MNIST validate the theoretical analysis, highlighting the trade-off between accuracy, forgetting efficiency, and storage cost, as well as the influence of unlearning request order.

Overall, the paper provides the first formal theoretical treatment of certified unlearning in continual learning, bridging two previously disjoint research areas and offering insights for privacy-preserving continual learning systems.

**Strengths:**

A substantive assessment of the strengths of the paper, touching on each of the following dimensions: originality, quality, clarity, and significance. We encourage reviewers to be broad in their definitions of originality and significance. For example, originality may arise from a new definition or problem formulation, creative combinations of existing ideas, application to a new domain, or removing limitations from prior results.

- Novel problem formulation: The paper introduces a new theoretical setting—continual learning–unlearning—that bridges two previously separate areas (machine unlearning and continual learning).
- Clear theoretical framing: It formalizes the trade-off between excess risk and unlearning loss, providing a unified analytical view and provable upper bounds.
- Extension of certified unlearning: The adaptation of gradient-based and Hessian-based certified unlearning algorithms to continual learning is technically interesting and nontrivial.
- Theoretical rigor: The derivations and proofs appear sound, grounded in convex optimization assumptions and differential privacy–style guarantees ((\varepsilon, \delta)).

**Weaknesses:**

A substantive assessment of the weaknesses of the paper. Focus on constructive and actionable insights on how the work could improve towards its stated goals. Be specific, avoid generic remarks. For example, if you believe the contribution lacks novelty, provide references and an explanation as evidence; if you believe experiments are insufficient, explain why and exactly what is missing, etc.

- Presentation issues: (1)Fig. 2 (a)(b)lack proper subfigure labels or axis annotations, making it difficult for readers to interpret what each curve represents. (2) Some notation (e.g., ( w^{-S_{1:t}}_t ), ( \gamma_t(S_{1:t}) )) is introduced without clear intuitive explanation, reducing readability.
- Limited practical insight: (1) The final conclusion is rather shallow; it restates the theoretical findings without extracting practical implications or design principles for real-world systems. (2)The theoretical results, while mathematically correct, have limited guidance for practical applications—for example, there is little discussion on how the proposed methods would scale to deep models or non-convex objectives.
- Clarity and exposition: Several sections (e.g., Algorithm 2 and its derivations) are dense and difficult to follow; figures or flow diagrams illustrating the unlearning process would improve clarity.
- Experimental limitations:  The evaluation is limited to MNIST with linear models, which does not convincingly support claims about the framework’s generality.
- Writing and structure:  The presentation could be reorganized to emphasize intuition before formal proofs. Some paragraphs in Section 3–5 are long and formula-heavy, which may discourage broader readership.

**Questions:**

Please list up and carefully describe any questions and suggestions for the authors. Think of the things where a response from the author can change your opinion, clarify a confusion or address a limitation. This is important for a productive rebuttal and discussion phase with the authors.

1. About prior work on unlearning without raw data:

    The authors claim that existing unlearning algorithms cannot handle cases where the original training data are unavailable. However, recent works such as *NoT: Federated Unlearning via Weight Negation* [Khalil et al., 2025] explicitly address this setting. The authors are encouraged to revise the statements in the *Introduction* to make them more precise and up to date.

2. Clarification on the second challenge (balancing two risks):

    The paper emphasizes the need to balance unlearning loss and the excess risk of continual learning to minimize post-unlearning excess risk. Could the authors explain more clearly why this trade-off inherently exists and what underlying mechanism causes these two objectives to conflict?

3. Motivation for unlearning in continual learning:

    The motivation for introducing explicit unlearning methods within continual learning needs clarification. Since deep neural networks already suffer from *catastrophic forgetting*, why is an additional unlearning mechanism necessary? What new privacy or system-level guarantees does this framework offer beyond natural forgetting?

4. Interpretability of key quantities:

    Equation (6) defines both *excess loss* and *unlearning loss* purely in mathematical form. A more intuitive explanation or visualization (e.g., what these losses mean in model behavior or output space) would greatly improve readability and help readers connect the theory to its practical implications.


[1]Y. H. Khalil, L. Brunswic, S. Lamghari, X. Li, M. Beitollahi, and X. Chen, “NoT: Federated Unlearning via Weight Negation,” arXiv.org. Accessed: May 19, 2025. [Online]. Available: https://arxiv.org/abs/2503.05657v1

---

### Official Review · Reviewer_QzyZ · 2025-10-31

**Soundness:** 2
**Presentation:** 2
**Contribution:** 1
**Rating:** 2
**Confidence:** 4

**Summary:**

This paper provides the first theoretical foundation for the important and novel problem of machine unlearning in a continual learning setting. Its main contribution is formalizing the problem and identifying the core trade-off between the "excess risk" from continual learning and the "unlearning loss." However, the work's impact is severely undermined by foundational assumptions that are incompatible with modern deep learning, methods with critical scalability issues, and experimental validation that is insufficient to support its claims.

**Strengths:**

1.  The paper's most significant strength is that it is the first to provide a rigorous theoretical framework for certified unlearning in the continual learning setting. By formally defining the objectives and constraints, it lays the groundwork and sets a clear research agenda for this important emerging field.
2. The decomposition of the post-unlearning excess risk is a key conceptual contribution. It moves beyond heuristics and provides a quantitative handle on the core trade-off between preventing catastrophic forgetting (which favors larger regularization) and enabling efficient unlearning (which, as the paper shows, favors smaller regularization). This is an elegant and powerful way to frame the problem.
3. The paper explores two distinct algorithmic pathways: the simple, storage-free "natural forgetting" approach and the complex, high-precision Hessian-based approach. Analyzing these two extremes provides valuable insights into the fundamental space-versus-accuracy trade-off that is central to this problem, offering a clear spectrum of potential solutions for future research.

**Weaknesses:**

1. On Theoretical Assumptions and Methodology:
•	The Strong Convexity Assumption is a Major Flaw: (Assumption 2.1) The entire theoretical framework relies on the assumption of µ-strong convexity. This is fundamentally misaligned with the non-convex reality of deep neural networks, making all derived bounds and guarantees largely inapplicable to the models of interest at ICLR.
•	Hessian-Based Method is expensive: Algorithm 2 requires storing a Hessian matrix (O(d^2)) for every task. This is computationally and memory-wise prohibitive for any non-trivial deep network, rendering the algorithm a theoretical exercise rather than a practical solution.
•	"Natural Forgetting" Method is Unreliable: Algorithm 1's performance may be problematic for recently learned tasks where natural forgetting is minimal. The resulting large approximation error would require injecting catastrophic levels of noise to ensure (ε, δ)-certification, likely destroying the model's utility.
•	Overly Simplified CL Model: The theory is based on a simple l2-regularization CL method, which is not representative of more effective, state-of-the-art CL approaches (e.g., replay-based methods). Conclusions drawn from this model may not generalize.
2. On Experimental Validation:
•	Experiments are Severely Limited: Validation was conducted only on MNIST using a linear model. This "toy" experiment is inadequate for a top-tier deep learning conference. It fails to demonstrate that the theoretical insights hold under the non-convex, high-dimensional dynamics of deep networks.
•	Lack of Meaningful Baselines: The paper only compares its two proposed methods. It lacks crucial comparisons to practical baselines, such as simple fine-tuning on the retained data or adapting existing static unlearning methods, which would be necessary to properly contextualize the methods' performance.
•	Superficial Analysis of Unlearning Sequence: The paper notes that the order of unlearning requests affects performance—an interesting finding—but provides no deeper analysis, theoretical explanation, or resulting guidance for algorithm design.

**Questions:**

1. The authors must explicitly acknowledge that the strong convexity assumption prevents the direct application of their theory to deep learning and discuss the implications. In addition, for the Hessian method to be considered viable, the authors should discuss and ideally experiment with scalable Hessian approximations (e.g., K-FAC, diagonal).
2. Can the authors improve the experiments on a standard vision dataset (e.g., CIFAR-10) with a real deep learning model (e.g., a small ResNet) to test if the theoretical trade-offs are still observed in a practical, non-convex setting?
3. To demonstrate value, the proposed methods must be benchmarked against simple yet relevant baselines like fine-tuning.

---

### Official Review · Reviewer_JRAS · 2025-11-01

**Soundness:** 1
**Presentation:** 2
**Contribution:** 2
**Rating:** 2
**Confidence:** 4

**Summary:**

The paper presents a theoretical framework for machine learning under continual learning settings. Under this framework, the authors present two methods: 1. A natural forgetting continual unlearning algorithm, and 2. A Hessian-based continual unlearning algorithm. Both algorithms rely on a EWC-style regularisation strategy to mitigate catastrophic forgetting. However, unlike EWC, the penalty is applied uniformly across all parameters without any Fisher/Hessian weighting. The two algorithms differ in their unlearning strategy.

The first algorithm adds isotropic Gaussian noise to the parameters with the standard deviation determined by a scalar influence bound that decays geometrically with the age of the deleted task. Upon the addition of noise, the model is published. An un-noised model is kept private for future training.

The second algorithm, after learning each task, caches the difference in model parameters and the full Hessian. During unlearning, a correction vector is built by pushing each deleted task’s update through the chain of later tasks with transport matrices computed from their respective Hessians. The model is then updated by adding the correction vector, and then noised and published.  To support their claims, the paper presents experiments on MNIST classification.

**Strengths:**

The problem is timely; the confluence of continual learning and unlearning is understood very little, and the paper attempts to build a theoretical angle to it, which is commendable.  The theories presented are extensive and backed up with proofs.

**Weaknesses:**

The theorems are presented under convexity assumptions. This need not hold true in practice, especially with deep learning models. The utility of unlearning lies primarily in large scale deep learning models where it is impractical to retrain without the forget-set.  It is also not clear if the proposed approach can scale to even moderately sized deep learning models.  The experiments are conducted with a linear model. The exact specifications of the model are not presented clearly.

Further concerning is the lack of comprehensive experiments to justify the claims presented empirically.

The paper presents a singular Split MNIST 3-way classification setup. This is a relatively simple problem. Stronger benchmarks are needed to properly demonstrate the effectiveness of continual learning and unlearning, particularly with regard to how well retained tasks are safeguarded from both catastrophic forgetting and unlearning.

The paper does not present accuracy numbers comparing task performance (both the target tasks being unlearnt and the others) before and after unlearning. This makes it hard to assess the effectiveness of the proposed method.

In Algorithm 1, the fact that noising is determined by a scalar bound that is not parameter-specific raises questions about the effectiveness of the proposed methods in terms of forgetting-spillover to the tasks that are to be retained.

Coming to Algorithm 2, the use of full Hessians is problematic in practice. Hessians are both expensive to compute and store, making the method impractical for anything beyond linear models. This could partly explain why the authors couldn’t present experiments in benchmarks more challenging than MNIST. One recommendation would be to study the use of approximations to the full Hessian, such as Kronecker-Factorized (K-FAC) Hessians and diagonal Hessians.

Further, caching model parameter differences adds to the storage cost. One could argue for a far simpler solution of having one model per task and simply discarding the model corresponding to the task to be unlearned. A similar approach is followed by Continual Learning and Private Unlearning by Bo Liu, Qiang Liu, and Peter Stone (https://arxiv.org/abs/2203.12817).

The lack of any baseline comparisons makes it hard to assess the proposed methods. Although Continual Learning and Unlearning is a relatively new topic, there exist works such as Continual Learning and Private Unlearning by Bo Liu, Qiang Liu, and Peter Stone (https://arxiv.org/abs/2203.12817), A Unified Framework for Continual Learning and Unlearning by Romit Chatterjee, Vikram Chundawat, Ayush Tarun, Ankur Mali and Murari Mandal (https://arxiv.org/abs/2408.11374) and An Unlearning Framework for Continual Learning
Sayanta Adhikari, Vishnuprasadh Kumaravelu and P. K. Srijith (https://arxiv.org/abs/2509.17530).

Under algorithm 1, unlearning new tasks could prove more problematic than unlearning older ones due to the decaying nature of influence. Newer tasks could be as likely, if not more,  to be unlearned as older tasks in many applications.

Under algorithm 1, storing a private model with data that is supposed to be unlearned could still be problematic under right-to-erasure compliance laws.

**Questions:**

The paper is titled “Machine Unlearning Meets Continuous Learning: A Theoretical Foundation”. In the literature, as well as in the rest of the paper, the term “Continual Learning” is used. Given the established nature of the term “Continual Learning,” it is curious why the authors chose an alternative.

In the Introduction (lines 51 & 52), the authors claim without reference that ChatGPT has adopted a continual learning approach. To the best of my knowledge, ChatGPT follows standard LLM pretraining protocols and doesn’t follow a continual learning protocol.

It would add to the paper’s clarity if the authors state that the Continual Learning problem that the paper chooses to address is Task Incremental Learning (TIL).

In Section 2.1, line 122 cites Elastic Weight Consolidation by Kirkpatrick et al. as assumption 2.1. There is a subtle difference in that Assumption 2.1 penalizes distance from the previous model, whereas EWC penalizes distance from the previous model, with the penalty weighted element-wise by the corresponding Fisher Information. While the effectiveness of EWC is widely known, that of assumption 2.1 is unclear. It would be beneficial if the authors attached a suitable reference that can attest to the effectiveness of the regularisation prescribed under Assumption 2.1.

The notation is confusing. In Section 2.2, line 133 states $S_t$ is a deletion request. Immediately after that, in line 134, $S_t$ is claimed to be a subset of task indices that have been trained but not deleted. It would improve readability if the passage could be rewritten.

---

### Meta-Review · Area_Chair_YcEE · 2026-01-05

**Summary:**

1. The reliance on the convexity assumption is validated only on a simple linear model, which is limited in scope.
2. The disconnect between unlearning's intended utility for large-scale deep learning and its limited testing on MNIST.
3. The computation of the full Hessian is impractical, especially for large-scale models.
4. Lack of comparisons with baseline methods.

**Reviewer Concerns:**

The authors did not provide a rebuttal.

**Reviewer Scores:**

The authors did not provide a rebuttal.

---

### Decision · Program_Chairs · 2026-01-26

Reject